# Cyclododecane-based high-intactness and clean transfer method for fabricating suspended two-dimensional materials

Zhao Wang [1,2], Wenlin Liu [1,3,4] ✉, Jiaxin Shao[4], He Hao[4], Guorui Wang[5], Yixuan Zhao[4], Yeshu Zhu[4], Kaicheng Jia[3], Qi Lu[6], Jiawei Yang[7], Yanfeng Zhang [2], Lianming Tong [3,4], Yuqing Song[2,3], Pengzhan Sun[8], Boyang Mao [9], Chenguo Hu [10] ✉, Zhongfan Liu [3,4] ✉, Li Lin [2,3] ✉ & Hailin Peng [3,4] ✉

The high-intactness and ultraclean fabrication of suspended 2D materials has always been a challenge due to their atomically thin nature. Here, we present a universal polymer-free transfer approach for fabricating suspended 2D materials by using volatile micro-molecule cyclododecane as the transfer medium, thus ensuring the ultraclean and intact surface of suspended 2D materials. For the fabricated monolayer suspended graphene, the intactness reaches 99% for size below 10 μm and suspended size reaches 36 μm. Owing to the advantages of ultra-cleanness and large size, the thermal conductivity reaches 2461 W m$^{-1}$K$^{-1}$ at 400.9 K. Moreover, this strategy can also realize efficient batch transfer of suspended graphene and is applicable for fabricating other 2D suspended materials such as MoS$_2$. Our research not only establishes foundation for potential applications and investigations of intrinsic properties of large-area suspended 2D materials, but also accelerates the wide applications of suspended graphene grid in ultrahigh-resolution TEM characterization.

Two-dimensional (2D) materials with atomic thickness exhibit various extraordinary properties compared to their bulk forms, and hold high promises in technological fields, including electronics, photonics and energy-related applications[1–4]. However, the properties of 2D materials are highly sensitive to the surroundings, such as the supporting substrates[5]. For example, the trapped charge and rough surface of silicon substrate strongly limit the electrical performance of graphene that resides on silicon substrate[6,7]. In this regard, suspended 2D materials can avoid the perturbation from substrates and allow for the approaching of intrinsic properties[8–11]. Furthermore, suspended graphene film with perfect hexagonal lattice, ultrahigh thermal/electrical conductivity, mechanical strength and negligible background noise has been proven to have great potentials in high-resolution transmission electron microscopy (TEM) and cryo-EM imaging which enables the construction of liquid cells[12,13], and atomic resolution of proteins[14–17].

However, high-intactness and clean fabrication of suspended 2D materials with sufficiently large free-standing regions remains

[1]College of Science, Northwest Agriculture & Forest University, Yangling, P. R. China. [2]School of Materials Science and Engineering, Peking University, Beijing, P. R. China. [3]Beijing Graphene Institute, Beijing, P. R. China. [4]Center for Nanochemistry, Beijing Science and Engineering Center for Nanocarbons, Beijing National Laboratory for Molecular Science, College of Chemistry and Molecular Engineering, Peking University, Beijing, P. R. China. [5]CAS Key Laboratory of Mechanical Behavior and Design of Materials, Department of Modern Mechanics, University of Science and Technology of China, Hefei, P. R. China. [6]College of Science, China University of Petroleum, Beijing, Beijing, P. R. China. [7]Faculty of Information Technology, Beijing University of Technology, Beijing, P. R. China. [8]Institute of Applied Physics and Materials Engineering, University of Macau, Avenida da Universidade, Taipa, Macau, P. R. China. [9]Department of Engineering, University of Cambridge, Cambridge, UK. [10]Department of Applied Physics, Chongqing University, Chongqing, P. R. China. ✉e-mail: liuwenlin@nwafu.edu.cn; hucg@cqu.edu.cn; zfliu@pku.edu.cn; linli-cnc@pku.edu.cn; hlpeng@pku.edu.cn

unachievable owing to the presence of generated cracks and contaminations. Conventionally, the fabrication of suspended 2D materials requires the transfer of 2D materials from growth substrates to holey substrates, such as TEM grid[12,13,15,18]. The polymers, such as polymethyl methacrylate (PMMA), are commonly used to continuously support the ultrathin films by suppressing the crack formation[19–21]. Without support from the underlying substrates, freestanding graphene films are prone to be torn by uncontrollable interfacial forces during the wet process. In this regard, the intactness of suspended area can be improved by replacing acetone with low surface tension solvent such as methoxynonaflurorbutane ($C_4F_9OH_3$) after the removal of PMMA[22–24].

Another key issue to be addressed in the preparation of freestanding graphene is surface cleanness, which is highly important in some applications, such as atomic resolution imaging. In traditional fabrication, the removal of supporting polymer is usually incomplete, thermal annealing at high temperature[25,26], and plasma treatment[27] are further utilized to remove residual polymer. However, severe conditions sometimes cause amorphous carbon and defects in graphene[28,29]. To resolve this problem, several polymer-free approaches have been reported through careful interfacial force control[9,13,15,18]. However, the free-standing region is still limited and can hardly reach over tens of micrometers. Therefore, sufficient support that would not contaminate 2D materials is fundamental for fabricating clean, high-intactness, and large-area free-standing 2D materials.

Here, we propose a universal transfer strategy to fabricate high-intactness and ultraclean suspended 2D materials by using nontoxic volatile cyclododecane (CD, $C_{12}H_{24}$) to support 2D materials. CD is firstly reported as a clean transferring medium by Capasso et al.[30], which can efficiently transfer CVD grown graphene to silicon wafer substrate[31–35], or onto indium tin oxide (ITO) glass substrate for solar cells applications[36]. In this work, focusing on high-quality suspended 2D materials fabrication, the complete and uniform sublimation of CD at room temperature can ensure a clean and intact surface. Besides, this strategy is applicable to various 2D materials, including graphene and $MoS_2$, and different holey goal substrates, such as TEM regular carbon grids, lacey carbon grids and SiN grids. Suspended graphene membrane was based on chemical vapor deposition (CVD)-derived graphene on Cu substrates, ensuring the scalability. The intactness is around 97% for 15 μm-sized freestanding region, and ~99% for suspended graphene with diameter below 10 μm, while the maximum suspended diameter for monolayer suspended graphene is up to 36 μm. Owing to the availability of ultraclean and free-standing graphene with large area, we further measured the intrinsic thermal conductivity of graphene and it is 2461 W m$^{-1}$ K$^{-1}$ at 400.9 K. We believe this efficient method for fabricating large-area clean suspended 2D materials can greatly promote the related investigation of intrinsic properties and also accelerate potential applications.

## Results

### The transfer of monolayer suspended graphene

CD is a stable cyclic hydrocarbon with non-toxic, eco-friendly and hydrophobic features. At room temperature, CD is a transparent solid, with high vapor pressure of 1.33 kPa at 100 °C and its melting point is around 60 °C. Owing to these characteristics, possible complete sublimation of CD ensures the availability of ultraclean surface. Figure 1a is the monolayer suspended graphene transferring process. After attaching TEM grid onto graphene assisted by evaporation of isopropanol (IPA), CD particles around 1 mg were introduced on graphene surface. CD was then melted through heating to fully cover graphene surface (Supplementary Movie 1), and it returned to solid state to form a stable compact supporting layer at room temperature. Particularly, CD can protect graphene membranes from the damages in the whole transfer process including aqueous etching, washing of

etchant, lifting graphene from aqueous etching solution, and drying. The detailed transferring process is shown in Methods.

The removal of CD can be achieved by its spontaneous sublimation. In the transferring, 6 h was required to completely remove CD at 40 °C under air pressure at room temperature. In addition, the strong mechanical properties of CD layer can sufficiently support graphene and the entire TEM array girds, enabling the transfer of large-area graphene onto over ten TEM grids at the same time to improve the production capacity (Supplementary Fig. 1), which is key to the commercial applications. Figure 1b shows the typical photograph of graphene on TEM holey gird covered by transparent CD, after drying of water. The monolayer and free-standing graphene is visible in the scanning electron microscopy (SEM) imaging. As evidenced by SEM image, graphene was successfully transferred onto TEM SiN grids with 5 μm and 10 μm-sized holes (Fig. 1c and Supplementary Figs 2 and 3) and high-intactness over 90% is obtained.

Higher intactness of free-standing graphene is achieved on TEM gird covered with holey amorphous carbon films or lacy carbon films owing to the stronger interaction between graphene membrane and flexible carbon film. Therefore, over 99% intactness is realized on monolayer graphene on TEM gird with 1.2 μm (Fig. 1d and Supplementary Fig. 4). Excitingly, near-100% intactness is achieved at some meshes with suspended size below 7 μm (Supplementary Figs 5–7), while over 98% intactness is possible for suspended region from 8 μm–15 μm (Fig. 1e and Supplementary Fig. 8). Supplementary Figs 9 and 10 exhibit the suspended graphene with large size in the range from 20 μm to 40 μm. With the assistance of the CD, we have achieved the efficient transfer of monolayer large-area graphene onto holey TEM grid, with a maximum free-standing region over 36 μm (Fig. 1f). Interestingly, with large suspended, graphene wrinkle would be formed and is clearly observed from the SEM images in Supplementary Fig. 11, which is presumably resulted from the tendency to reduce the surface energy. In contrast, with smaller hole size, almost no wrinkle would be formed. Moreover, the intactness of free-standing graphene as function of suspended size below 15 μm is carefully calculated, with corresponding statistic presented in Fig. 1g. For the suspended diameter larger than 15 μm, the statistical intactness result cannot be calculated accurately because the corresponding total number of the large hole in one grid in commercially available TEM lacy girds is commonly very limited. The above observation strongly confirms the capability of our strategy for fabricating large-area, atomically thin, free-standing 2D materials with no observed contaminations in SEM image.

### Clean transfer with CD

To probe the surface cleanness of the as-received free-standing graphene, we further carried atomic force microscopy (AFM) and TEM characterization. Firstly, after the sublimation of CD, optical microscopy (OM) image of graphene membrane transferred to Si/SiO$_2$ substrate exhibits a uniform contrast (Fig. 2a), and X-ray photoelectron spectroscopy (XPS) spectrums in Supplementary Fig. 12 indicate no observable Na and Cu elements on graphene. Besides, the AFM images with little contaminations and a very low roughness also confirm the clean surface after transfer (Fig. 2b, c). From some typical AFM images (Supplementary Fig. 13), the impurity residue is around 1.2% by calculating the area ratio of impurity-occupied region to the entire graphene films (Supplementary Fig. 14), which may origin from residual copper oxide and airborne contaminations. In addition, one important thing should be noted that it is hard to achieve good contact between silicon wafer substrate and 2D films because CD is quite rigid. While in fabricating suspended 2D thin films on TEM holey grid, CD can bond 2D thin films with TEM gird well together through the holes on TEM grid. This bonding ensures the good contact between 2D thin films and TEM substrate to achieve high yield and high-intactness suspended 2D materials. AFM characterization of suspended graphene on TEM holey

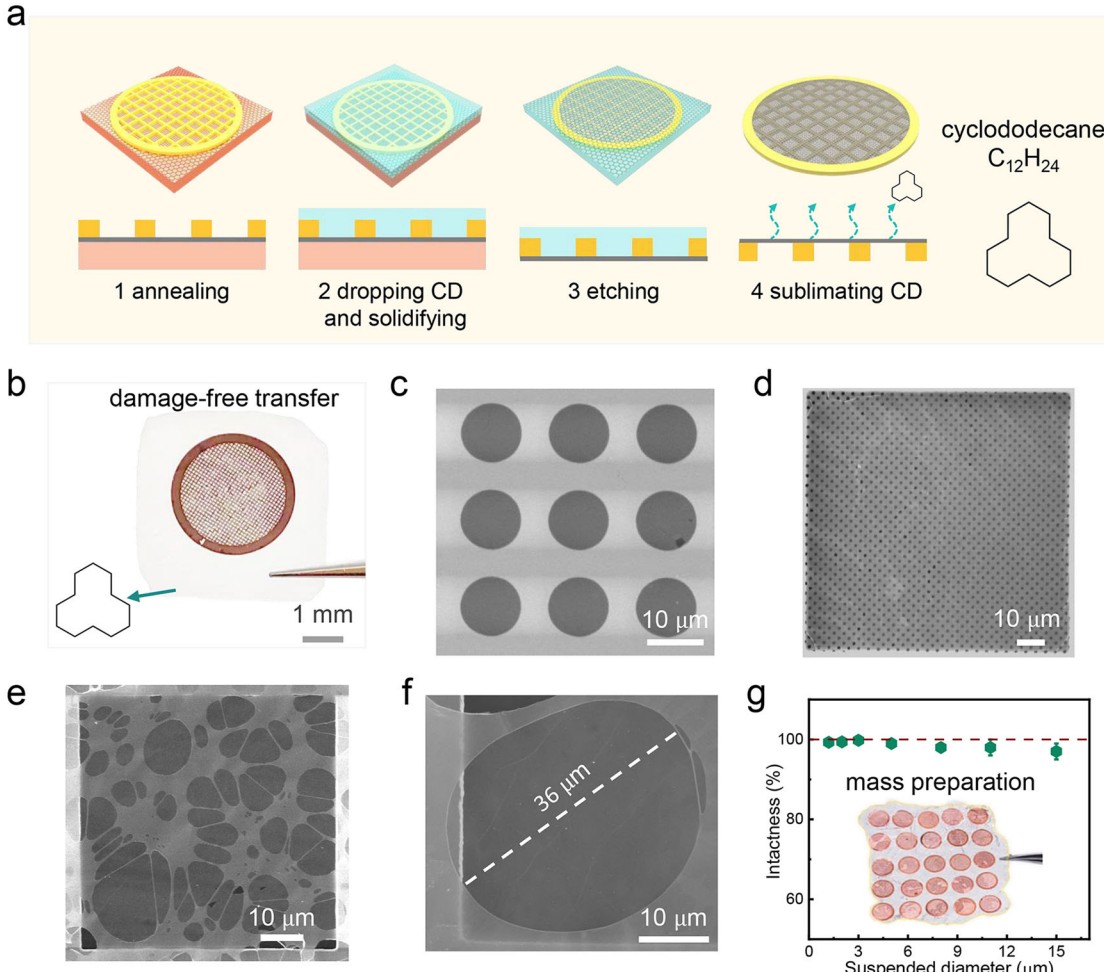

**Fig. 1 | High-intactness and clean transfer method for single-layer suspended graphene with cyclododecane (CD) as support layer. a** Schematic illustration of the procedure for transferring graphene onto transmission electron microscopy (TEM) holey substrate. **b** The optical image of the as-transferred graphene on TEM grid with the assistance of CD. The light transparent substance is CD. **c** Scanning electron microscopy (SEM) image for single-layer suspended graphene on SiN holey substrate, the diameter for each hole is 10 μm. **d** Suspended graphene on TEM carbon substrate with uniform hole distribution, the diameter for each hole is 1.2 μm and the intactness is about 99%. **e** Suspended graphene on TEM lacey carbon substrate with holes in roughly 8 μm average diameter, and the calculated intactness is about 99%. **f** The suspended diameter for the single-layer suspended graphene is about 36 μm. **g** Intactness at different suspended diameters for single-layer suspended graphene obtained in this work. The error bar is obtained by the statistical results in Supplementary Figs 4–8.

substrate also supports the absence of contaminations (Fig. 2d). In the TEM image of suspended graphene (Fig. 2e), only amorphous carbon originated from the high-temperature growth was observed with preserved graphene hexagonal lattice, verified by the selected area electron diffraction (SAED) patterns (Fig. 2f).

The preparation of suspended graphene membrane is capable of excluding the doping and strain effect from substrates. In this regard, Raman characterization was performed to confirm the intrinsic properties of as-received graphene. As displayed in Fig. 3a, b and Supplementary Fig. 15, a high ratio of the intensity of 2D peak ($I_{2D}$) to the intensity of G peak ($I_G$) and no defect-related D peak of the suspended graphene transferred by CD confirm reduced doping and strain level in comparison with supported graphene and suspended graphene transferred by PMMA. The full width at half-maximum of 2D peak ($FWHM_{2D}$) is also reported to be an indicator of strain and doping[37]. As presented in Fig. 3c, the statistical $FWHM_{2D}$ of suspended graphene films in different TEM grids exhibits a low value of around 22 cm$^{-1}$, confirming the improved quality owing to the clean surface free of additional doping and strains. Mobility is another effective way to reflect the performance of suspended graphene. Generally, the suspended device is fabricated by depositing electrodes on suspended graphene (method 1) or transferring both deposited electrodes and

graphene onto a hole substrate (method 2). However, using CD as transfer medium, we must acknowledge the big challenges in fabricating device array without any damage to suspended single-layer graphene, (a) inevitable damage to suspended graphene in depositing electrode process for method 1, (b) the poor contact between TEM grid and graphene resulting from the additional thickness of the deposited electrodes for method 2, (c) difficulty in precisely locating the patterned suspended regions determined by TEM grids and (d) easily destruction by the external force in introducing mental needles for testing. Fortunately, $FWHM_{2D}$ can link to the carrier mobility of device[38]. Raman mapping result for $110 \times 110 \, \mu m^2$ intact suspended area (Supplementary Fig. 16 and Supplementary Table 1) and the estimated carrier mobility in Supplementary Fig. 17 further reflect the high large-area uniformity of electronic quality of suspended graphene obtained in this work. Furthermore, the plot of peak position of 2D as function of G peak position also confirms the near-intrinsic nature of free-standing fabricated by CD (Fig. 3d).

**Thermal conductivity of large clean suspended graphene**
In addition, we also measured the thermal conductivity ($\kappa$) of suspended graphene based on Raman methods. In general, in the case of graphene, the thermal conductivity is sensitive to the crystal quality

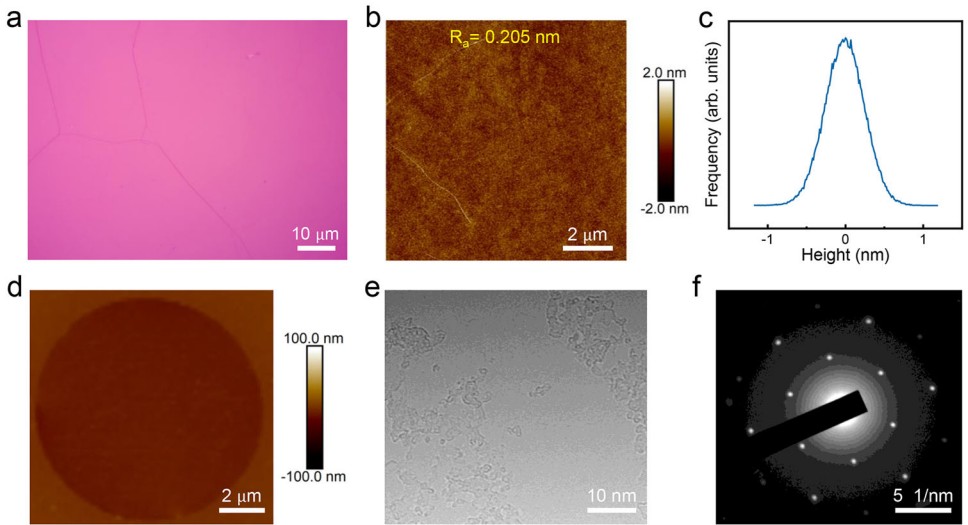

**Fig. 2 | Clean transfer method with CD support layer. a** The optical image of single-layer graphene transferred onto Si wafer with the assistance of CD. **b**–**c** Atomic force microscopy (AFM) characterization and the height distribution of graphene transferred onto Si wafer by CD. Ra: average roughness. **d** AFM characterization of the fabricated single-layer suspended graphene on SiN holey substrate using CD as support layer, the diameter for the suspended region is 10 μm. **e**–**f** TEM characterization and selected area electron diffraction (SAED) of the transferred single-layer suspended graphene on SiN holey substrate.

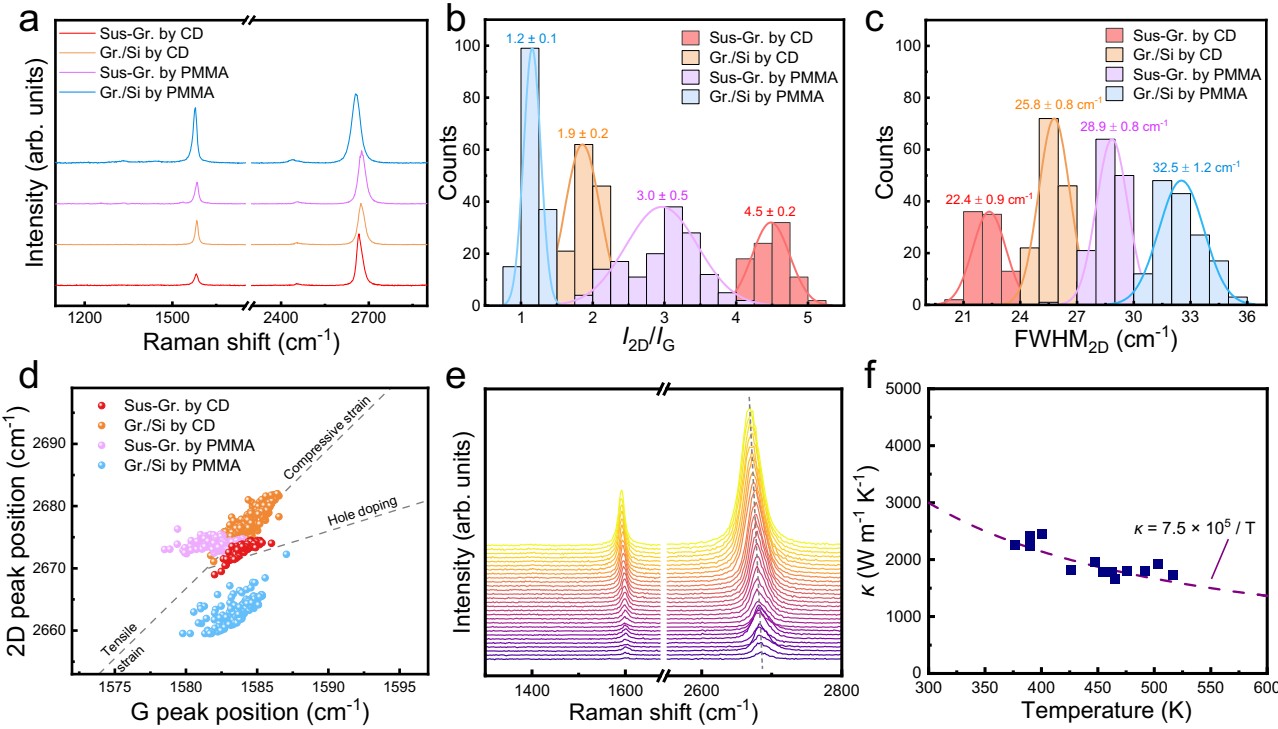

**Fig. 3 | High quality single-layer suspended graphene. a** Raman spectra of suspended graphene transferred by CD (Sus-Gr. by CD in red) or by polymethyl methacrylate (PMMA) (Sus-Gr. by PMMA in purple), and supported graphene on Si wafer transferred by CD (Gr./Si by CD in orange) or by PMMA (Gr./Si by PMMA in blue). **b** Statistical result of the intensity ratio of Raman 2D to G peaks ($I_{2D}/I_G$). **c** Statistical result of the peak width at half height of 2D peaks (FWHM$_{2D}$). The solid lines in (**b**) and (**c**) are the fitted distribution curves. **d** The relationship of 2D peak position and G peak position from Raman spectra. **e** Raman spectra of single-layer suspended graphene film excited at powers from 0.2 mW to 11.0 mW. The measured suspended single-layer graphene is on Au-coated SiNx holey grid with 10 μm suspended diameter. **f** The calculated thermal conductivity as a function of temperature (purple points) with a fit (the dashed line).

and surface cleanness[39–42]. In the thermal measurement, through × 50 objective lens, a laser beam with 532 nm wavelength was focused on the center of the single-layered suspended graphene on Au-coated SiNx grid with 10 μm-sized hole. Thus, heated by the laser, Raman of graphene exhibited a clear red shift owing to the increased anharmonic scattering of optical phonons. Based on the detailed thermal diffusion mode, thermal conductivity can be obtained by the follow calculation[43–45]:

$$k = \frac{\ln\frac{R}{r_0}}{2\pi d\frac{T_m - T_a}{Q - Q_{air}}}\alpha \tag{1}$$

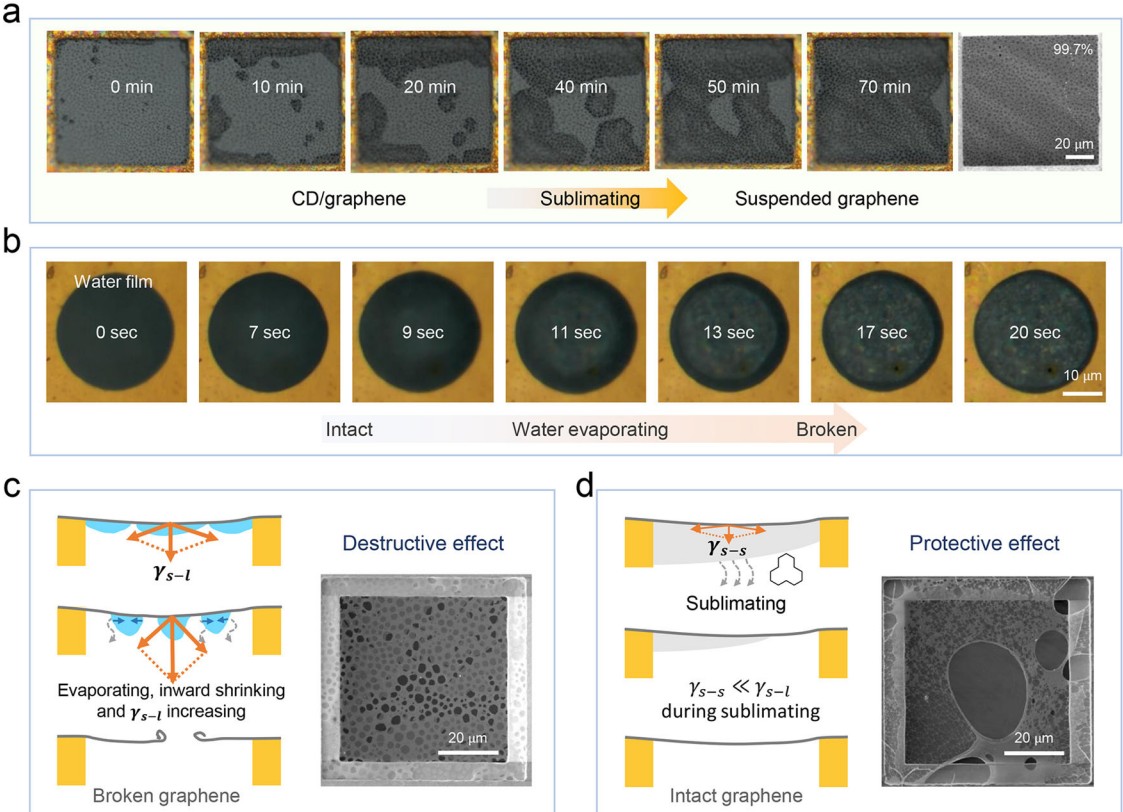

**Fig. 4 | The mechanism for transferring high-intactness and clean suspended graphene membrane with CD as support layer. a** Optical image of the as-transferred graphene on TEM holey lacey grid with the assistance of CD after completely etching Cu substrate, we can acquire the sublimation way of CD from the images at different time stages. The rightest picture is the SEM image of the final obtained single-layer suspended graphene and the intactness is quite high. **b** The dry process of suspended water membrane on TEM SiN holey substrate, it is observed that continuous water membrane can support graphene while fractured water membrane cannot do that and the water membrane can be broken within several seconds. **c** Illustration of dry process in traditional transfer without CD, and single-layer suspended graphene with size larger than 5 μm can be easily broken. $\gamma_{s-l}$ is the interfacial tension between graphene and liquid. **d** Illustration of sublimation of CD in our transfer way with CD as the support layer, and single-layer suspended graphene films with quite large size can be obtained. $\gamma_{s-s}$ is the interfacial tension between graphene and CD.

Where $R$ is the radius of hole, $r_0$ is the radius of the Gaussian laser beam, $\alpha$ is the Gaussian profile factor of the laser beam, and $d$ is the thickness of graphene. In this work, $R$ is 5 μm, $r_0$ is 0.226 μm for × 50 objective lens, $d$ is 0.335 nm for monolayer graphene, and $\alpha$ equals 0.98. In addition, $T_m$ is the measured temperature, $T_a$ is ambient temperature which equals to 300 K, $Q$ is absorbed laser power, while $Q_{air}$ is power loss to air during the laser heating.

$T_m - T_a / Q - Q_{air}$ represents the increased temperature at the center of the suspended graphene caused by the absorbed laser power, which can be deduced by matching the calculated temperature rise ($Q - Q_{air}$, Fig. 3e) with the temperature rise measured by Raman spectroscopy ($T_m - T_a$, Supplementary Fig. 18a). The detailed 2D peak positions at different laser powers are summarized in Supplementary Fig. 18b. Thus, the thermal conductivity $\kappa$ can be extracted to 2461 W m$^{-1}$ K$^{-1}$ at 400.9 K (Fig. 3f), displaying a high thermal performance and near-intrinsic properties of suspended graphene.

**Mechanism behind the high-quality transfer**

To understand the contribution of CD to the observed high-intactness transfer, we monitor the temporal evolution of the CD on the graphene surface during the spontaneous sublimation (Fig. 4a). Directly after the drying of water, the graphene was still fully covered by the CD, indicating the role of CD in protection of graphene against crack formation. Exposed to air, the sublimation was observed to be faster at the edge of TEM grids, which usually occurred from the edge to the center. As illustrated in the corresponding OM images and the corresponding

Supplementary Movie 2, after roughly 70 min under light, the sublimation of CD was complete, leaving free-standing graphene intact. And it should be noted that the complete sublimation time for CD is much longer at room temperature.

In contrast, during the drying of water from the free-standing graphene film without CD (Fig. 4b and Supplementary Movie 3), the water film can only sufficiently support 2D materials by fully covering the membranes at the early stage. Upon the evaporation of water at some region, the water layers would shrink, producing surface tension that is responsible for the crack formation even within 20 s. It has been reported that the surface tension of water is measured to be approximately 72.8 mN m$^{-1}$ at room temperature[46]. Such a high surface tension of water would cause a remarkable stress of graphene ($\gamma_{s-l}$) especially for single-layer graphene whose stress is the largest comparing to few-layer graphene, and the catastrophic rupture can be expected once the fracture strength of graphene is reached[14]. Furthermore, water tends to converge to both sides of graphene surfaces during the drying process. In such case, with the higher volume of concentrated water, the more stretching of graphene and the higher risk of fracture would be expected (Fig. 4c). Therefore, without using CD, the drying process of water would result in the cracked graphene films.

Here, CD plays a protection role for graphene films. This is because on one hand CD has a lower surface energy comparable to a hydrocarbon wax, around 30.6 mN m$^{-1}$ at room temperature[47,48]; The lower surface energy would contribute to the alleviating of the

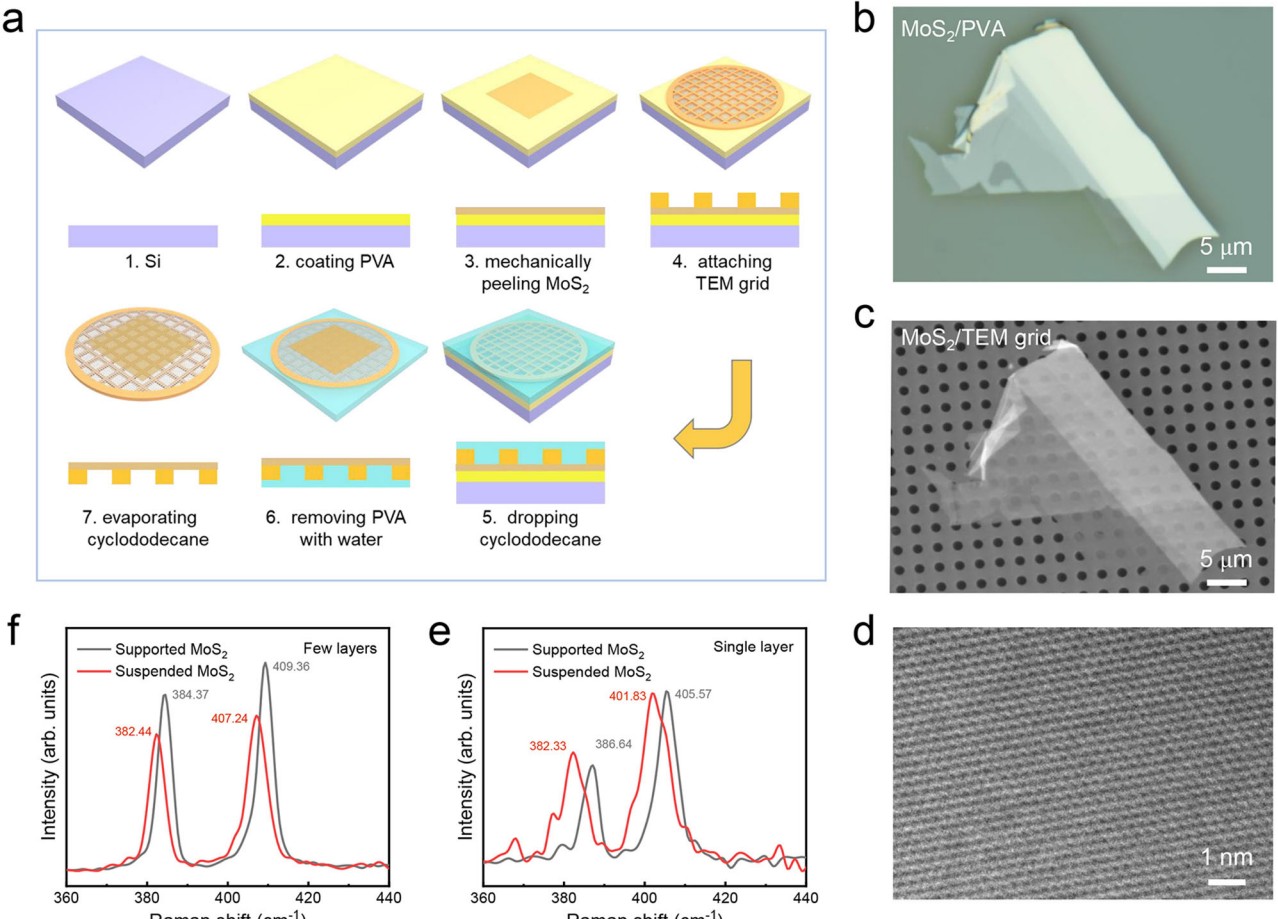

**Fig. 5 | Fabrication of suspended MoS₂ films. a** Schematic illustration of the procedure for transferring mechanically peeled MoS₂ flakes onto TEM holey substrate with CD as the support layer. **b** The optical image of mechanically peeled MoS₂ on polymer polyvinyl alcohol (PVA) substrate, corresponding to the third stage in the transferring procedure in (**a**). **c** SEM image of the final obtained suspended MoS₂ flakes on TEM holey grid. **d** High-magnification TEM images of suspended MoS₂ flakes. **e**–**f** Raman characterization of supported/suspended MoS₂ in single layer and few layers, respectively.

stretching in graphene ($\gamma_{s-s}$) and thereby suppress the crack formation. From the mechanical compression curve of CD, the corresponding Young's modulus is calculated to be 102.8 Mpa, such a strong mechanical property of CD ensures sufficient support from the compact CD layer to graphene for a large crack-free suspended area (Supplementary Fig. 19). In addition, we note that CD experiences a mild sublimate upon air exposure which was completed with relatively uniform thinning process. It can be envisaged that such relatively uniform thinning of CD layer without aggregation circumvents the stress concentration issues, ensuring the strain level in graphene is below the fracture limit[49], as demonstrates in Fig. 4d. Thus, CD is an ideal support layer during its removal by sublimation. Besides, note that in our strategy, the intimate contact between graphene and TEM holey substrate achieved through thermal annealing is also key to the final intactness of the suspended graphene (Supplementary Fig. 20).

### Fabrication of other free-standing 2D materials

Furthermore, we extended the transfer method to other 2D materials: transferring mechanically exfoliated MoS₂ and CVD-grown MoS₂ films onto TEM holey substrate to fabricate free-standing MoS₂ films. The specific process for transferring mechanically exfoliated MoS₂ is demonstrated in Fig. 5a. During the transfer, we first spin-coated a water-soluble polymer polyvinyl alcohol (PVA) layer on Si substrate. Next, MoS₂ flakes were exfoliated onto PVA layer (see "Methods" for details). After the attachment of TEM holey grid on MoS₂ flakes with the assistance of IPA, CD was introduced onto TEM grid. The whole stack was subsequently immersed into deionized water to remove PVA. After the washing and sublimation of CD, suspended MoS₂ was successfully fabricated. Different from transferring mechanically exfoliated MoS₂, transferring CVD-grown MoS₂ to fabricate suspended MoS₂ films requires polymethyl methacrylate (PMMA) and thermal release tape (TRT) to peel off MoS₂ from sapphire grown substrate in NaOH solution before coating PVA layer (see Supplementary Fig. 21 and Methods for details).

Figure 5b is OM image of the mechanically exfoliated MoS₂ flakes on PVA layer with different contrasts shown, illustrating the different thickness of MoS₂ flakes. After transferring, the SEM image of the suspended MoS₂ flakes on TEM holey grid in Fig. 5c displays the same morphology with Fig. 5b, illustrating the high-intactness of the transferred MoS₂. Besides, for the fabricated suspended MoS₂ transferring from sapphire grown substrate, SEM image in Supplementary Fig. 22 displays large discontinuous films composed of triangular morphologies of MoS₂ domains, which also proves the high-intactness transfer with the assistance of CD. High-resolution TEM image of suspended MoS₂ transferring from exfoliated flakes (Fig. 5d) displays a clean surface with clean lattice structure of MoS₂. As indicated in the corresponding Raman spectrum in Fig. 5e, f, transferred MoS₂ flakes display two characteristic modes, $E_{2g}^{1}$ and $A_{1g}$ modes, belonging to in-plane and out-of-plane vibrations of Mo and S atoms, respectively. And 19 cm⁻¹ peak frequency separation confirms the monolayer nature of the MoS₂, while larger peak frequency separation corresponds to multilayer MoS₂[50–53]. Thickness-dependent Raman spectrums with

different layer numbers transferring from exfoliated MoS$_2$ flakes and for single-layer suspended MoS$_2$ transferring from sapphire substrate are presented in Supplementary Figs 23 and 24, respectively. Besides, the Raman spectra of supported MoS$_2$ on PVA polymer and suspended MoS$_2$ on TEM holey grid are both measured. A red shift is observed in the two vibrational modes for suspended MoS$_2$, which can be attributed to the release of strain in the suspended structure[54]. In addition, the intensity ratio of $E_{2g}^1$ to $A_{1g}$ for suspended MoS$_2$ is larger than that of supported MoS$_2$, presumably owing to the increased out-of-plane motions of S atoms in the free-standing modes[53].

## Discussion

We have developed a strategy to fabricate suspended 2D materials with high-intactness and clean surface relying on the use of CD as the transferring media, which is non-toxic and can be completely removed by self-sublimating. Monolayer suspended graphene films were successfully fabricated on various TEM holey substrates. The suspended diameter is up to 36 μm, and the intactness is higher than 98% for suspended diameter below 15 μm. The clean surface of suspended graphene contributes to a high thermal conductivity of 2461 W m$^{-1}$ K$^{-1}$. Suspended MoS$_2$ was also successfully transferred with the assistance of CD. This work not only provides an efficient method for preparing large-area suspended 2D materials for potential applications and investigation of intrinsic properties, but also accelerates the wide researches and applications of high resolution TEM characterization based on suspended graphene.

## Methods

### Preparation of single-crystal Cu (111) wafer on sapphire

Single-crystal Cu was prepared through deposition on single crystal sapphire wafers. Before deposition, sapphire wafer was annealed in a pure oxygen atmosphere at 1020 °C for 6 h under 1 atm pressure to ensure the quality of single-crystal Cu[55]. After that, Cu film with roughly 500 nm thick was deposited on the sapphire wafer by radio frequency sputtering equipment. Then the Cu/sapphire sample was annealed in 1000 standard cubic centimeters per minute (sccm) Ar and 100 sccm H$_2$ atmosphere at 1020 °C for 2 h under 1 atm pressure.

### Growth of graphene films on Cu/sapphire wafers

Before the growth of graphene films, Cu (111)/sapphire wafer was first heated to 1020 °C in an atmosphere-pressure CVD system with 1000 sccm Ar and 100 sccm H$_2$. Subsequently, 100 sccm CH$_4$ (0.1% diluted in Ar) was introduced to initiate the growth of the graphene, and graphene wafer with full coverage could be obtained after 2 h growth. After graphene growth, the system was cooled down to room temperature under the same gas flow.

### Growth of graphene films on Cu foil

The graphene films grown on Cu foil were using the low-pressure CVD system with quartz tube furnace. The Cu foil with 50 μm thick that was purchased from Kunshan Luzhifa Electronic Technology Co., Ltd was used as the growth substrate. Before the growth of graphene film, Cu foil was first heated to 1020 °C with 500 sccm Ar and then is annealed with 500 sccm H$_2$ for 30 mins. Subsequently, introducing 1 sccm CH$_4$ to grow continious graphene films for 1 h. After that, the system was cooled down to room temperature under the same flow.

### Transferring graphene procedures to Si wafer

Both graphene films on Cu/sapphire wafer and Cu foil-grown substrates can be transferred to Si wafer. The transferring operation was performed on a clean bench. Firstly, the graphene on the back side of Cu foil was removed by Plasma etching an oxygen atmosphere, and then solid CD particles (purchased from Macklin, ≈ 2 mg) were introduced onto the selected area of graphene film using clean tweezers. Next, the CD was melted at 70 °C on a hot plate for several minutes, allowing it to flow and fully cover the surface of the graphene film. Afterward, the sample was removed from the hot plate and allowed to cool down to room temperature, causing CD to return to its solid state and firmly attach to the graphene. Cu substrate was then etched with 0.5 M Na$_2$S$_2$O$_8$ solution, followed by washing with deionized water for several times. The CD/graphene was then taken out of water and placed to clean Si wafer. Finally, CD was completely removed by sublimating at 40 °C under air pressure for roughly 6 h or longer, and graphene on Si wafer was obtained.

### Highly efficient fabrication procedures for monolayer suspended graphene films

Graphene films grown on Cu/sapphire wafer and Cu foil grown substrates can be both used to fabricate suspended graphene. Various types of holey TEM grids can be used as the supporting substrates for fabricating suspended graphene, such as TEM silicon nitride grids with uniform hole size, TEM grids with uniform holey amorphous carbon film and TEM lacey grids with holey amorphous carbon film in different hole sizes.

For fabricating suspended graphene, firstly, selecting a certain area of graphene films on growth substrate with comparable size to that of TEM grid. Then direct pressing TEM grid down onto graphene, IPA solution was subsequently dropped to the sample and the evaporation of IPA would ensure enhanced interaction between graphene and TEM grid. Thereafter, TEM grid/graphene/Cu was annealed at 250 °C for 2 h. After that, solid CD particles roughly at 1 mg were introduced by tweezers onto the surface of the prepared sample at room temperature. Then, CD/TEM grid/graphene/Cu sample was heated at 70 °C on hot plate to initiate the melting of CD, which would cover the entire surface of graphene. Subsequently, the cooling of the sample would solidify CD again, followed by the etching of the Cu substrate using Na$_2$S$_2$O$_8$ solution (0.5 M). Magnetic stirring is required to accelerate the etching speed especially in transferring graphene gown on Cu/sapphire wafer. After Cu substrate was completely etched away, the CD/TEM grid/graphene sample was carefully cleaned by deionized water for several times to remove metal ions. Finally, after drying of water and complete removal of CD by sublimating at 40 °C at air pressure, monolayer suspended graphene films were obtained.

### Fabrication procedures for suspended MoS$_2$ films transferring from mechanically peeled MoS$_2$

Firstly, polyvinyl alcohol (PVA) with molecular weight of 31000 purchased from adamas-beta. com was dissolved in pure water with 10 wt% under magnetic stirring at 70 °C for several hours. Then, a thin layer of PVA was spin coated onto Si wafer at 1000 rpm for 10 s and followed by heated at 60 °C for 2 h. Subsequently, MoS$_2$ flakes were attached on PVA layer gently via blue tape (Nitto Denko). After removing blue tape, TEM holey substrate was attached to the selected area of MoS$_2$ flakes through delicate operation under optical microscope. The prepared TEM grid/MoS$_2$/PVA/Si sample was then heated at 60 °C to stick TEM grid on PVA layer. A very little IPA solution was introduced to enhance the interaction. Subsequently, solid CD particles roughly at 1 mg were added on the sample. and CD was melted at 70 °C to fully cover the surface of sample. After CD returned to solid state at room temperature, PVA layer was removed by washing with deionized water for several times. And finally, CD was gradually sublimated by heated at 40 °C under air pressure, and suspended MoS$_2$ film on TEM holey gird was successfully obtained.

### Fabrication procedures for suspended MoS$_2$ films transferring from CVD-grown MoS$_2$

The transferred MoS$_2$ thin film was grown on sapphire substrate. First, polymethyl methacrylate (PMMA, average Mw ≈ 910 000, Macklin) was dissolved in anisole with 4 wt% under magnetic stirring at 60 °C for several hours. This solution was subsequently spun onto CVD-grown

$MoS_2$ at 1000 rpm for 60 s, repeated twice. Afterward, the sample was heated at 60 °C for 5 min to form a PMMA thin layer. Then, thermal release tape (TRT, 3198MS, Nitto Denko) was attached onto PMMA layer. After immersing the whole stack in NaOH solution (1 M) for 5 min, $MoS_2$ film was delaminated from sapphire. Thereafter, the prepared $MoS_2$/PMMA/TRT was washed in deionized water for several times. After the sample became dry, 10 wt% PVA was spin coated on $MoS_2$ layer at 1000 rpm for 60 s, repeated twice. And the sample was then heated at 60 °C for 5 min. After that, polyethylene terephthalate (PET) layer was attached on PVA layer. Then, $MoS_2$ layer was directly peeled off from PMMA layer. Holey TEM grid was then attached on $MoS_2$/PVA/PET, and a little IPA solution was dropped to enhance the interaction between TEM grid and $MoS_2$. Subsequently, solid CD particles ($\approx$ 1 mg) was introduced onto TEM grid by tweezers. After CD melted at 70 °C, the whole stack was fully immersed in deionized water to remove PVA. Finally, after the complete CD sublimation at 40 °C - 50 °C under air pressure, suspended $MoS_2$ on TEM grid was successfully fabricated.

### Characterization

The morphology and structure of the fabricated suspended graphene or suspended $MoS_2$ were characterized with SEM (Hitachi S-4800, acceleration voltage 5 kV), atomic force microscopy (AFM) (Bruker dimension icon, scansyst mode, scansyst air tip), and Raman spectrometer (Horiba, LabRAM HR-800, 532 nm laser wavelength, × 100 objective). For measuring the Raman spectrum of suspended $MoS_2$, the power of the excitation laser is 1 mw to minimize sample damage. TEM characterization of the suspended samples is carried by TEM (FEI Tecnai T20, acceleration voltage 200 kV). Thermal conductivity experiments of monolayer suspended graphene on Au-coated SiNx grid have been carried out at room temperature using the 532 nm wavelength as the excitation source and a confocal micro-Raman setup (Witec Alpha-300) equipped with a × 50 micro-scope objective. The laser power passing the objective lens was measured by an optical power meter. A heating stage was used for temperature control in Raman characterization.

### Data availability

The Source Data underlying Figs. 3, 5e-f and Supplementary Figs. 12, 15, 16d, 18, 19a, 23 and 24 are available at https://doi.org/10.6084/m9.figshare.25746207. All raw data generated during the current study are available from the corresponding authors upon request.

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

## Acknowledgements

This work was financially supported by the National Key Research and Development Program of China (2022YFA1204900, L.L. and H.P.), the Talent Foundation of Northwest Agriculture & Forest University (Z1013023003, Z.W., Z1013023002, W.L.), the National Natural Science Foundation of China (52372038, L.L., T2188101, Z.L., 52021006, H.P., 12202430, G.W., 12241202, G.W. and 22105006, Y.S.), and the Fundamental Research Funds for the Central Universities (2452024094, Z.W., 2452024090, W.L.). The authors acknowledge the Molecular Materials and Nanofabrication Laboratory (MMNL) in the College of Chemistry at Peking University for the use of instruments.

## Author contributions

L.L., W.L., Z.L., and H.P. supervised the project. Z.W and W.L. carried the suspended graphene and $MoS_2$ transferring part and took and analyzed optical microscope, SEM, AFM, and XPS data. Z.W. carried Raman measurements under different temperatures, and Z.W. and H.H. carried Raman measurements under different powers. J.S. conducted the annealing process. L.L., W.L., and G.W. completed the mechanism analysis. X.Y.Z. took the TEM images. S.Y.Z. and K.J. conducted the CVD growth of graphene films. Q.L., J.Y., and Y.S. gave suggestions on suspended graphene transferring. F.Y.Z. provided CVD-grown $MoS_2$. L.T., P.S., B.M., and C.H. gave suggestions on the experimental data analysis. Z.W., W.L., and L.L. wrote the manuscript.

## Competing interests

The authors declare no competing interests.
