## [Peer Review File · Nature Communications]

Cyclododecane-based high-intactness and clean transfer method for fabricating suspended two-dimensional materialsEditorial Note: Parts of this Peer Review File have been redacted as indicated to remove third-party material where no permission to publish could be obtained.

REVIEWER COMMENTS

Reviewer #1 (Remarks to the Author):

In this paper, the authors demonstrate the use of cyclododecane as a transfer medium for the fabrication of suspended graphene and other 2D material membranes with high yield. AFM topography mapping, TEM and Raman mapping analysis has been used to demonstrate the cleanliness and low built-in strain and doping of the suspended graphene. Raman is also used to confirm the high thermal conductivity which can be linked to the cleanliness.

CD was introduced as a sacrificial transfer medium for graphene 10 years ago, but this paper does not acknowledge the literature that exists, for instance, Appl. Phys. Lett. 105, 113101 (2014) and Brajpuriya, R. JMR 36 (2021). While this was not for transferring suspended membranes, nonetheless it represents important prior art for the use of CD for graphene transfer and must be noted explicitly.

The authors themselves have previously published the transfer of suspended graphene on TEM grids (without the use of CD), but these references (12, 13) are buried in the first generic sentence and not explicitly noted. This is particularly significant because the phenomenon described in fig 4 of this paper was already noted in Ref 12, but there is no explicit reference to this past work. This should be corrected.

Noting the above omissions, this work combines the previous work from the authors on suspended graphene transfer with the use of CD which has been demonstrated for graphene transfer before, to fabricate high-yield large-diameter suspended graphene membranes. This does indeed represent a significant advancement in the state of the art and worthy of publication.

There are numerous language errors in this manuscript and the authors are encouraged to undertake professional language revision of this paper to ensure a high language standard prior to publication.

Reviewer #2 (Remarks to the Author):

The paper “*High-intactness and clean transfer for fabricating suspended two-dimensional materials*” from Wang et al. focuses on the cyclododecane-assisted method to transfer CVD graphene. The method has been proposed for few-layer graphene in 2014 by Capasso et al. and is well known in the 2D materials community (the original paper has more than 60 citations - **Cyclododecane as support material for clean and facile transfer of large-area few-layer graphene**, [10.1063/1.4895733](https://doi.org/10.1063/1.4895733)). In 2023, the same group proposed a refinement of the cyclododecane-assisted method to afford large-area transfer of monolayer graphene (**Solvent-free transfer of monolayer graphene with recrystallized cyclododecane**, [10.1063/5.0169748](https://doi.org/10.1063/5.0169748)).

(Incidentally, these are some more papers, among the others, utilizing the cyclododecane-assisted method:

Nitrogen-doped graphene films from chemical vapor deposition of pyridine: influence of process parameters on the electrical and optical properties [10.3762/bjnano.6.206](https://doi.org/10.3762/bjnano.6.206)

Chemical Vapor Deposited Graphene-Based Derivative As High-Performance Hole Transport Material for Organic Photovoltaics [10.1021/acsami.6b06749](https://doi.org/10.1021/acsami.6b06749)

Contamination-free graphene by chemical vapor deposition in quartz furnaces 10.1038/s41598-017-09811-z

Ethanol-CVD Growth of Sub-mm Single-Crystal Graphene on Flat Cu Surfaces [10.1021/acs.jpcc.8b10094](https://doi.org/10.1021/acs.jpcc.8b10094)

In the submitted paper, Wang et al. used the cyclododecane-assisted method to prepare suspended graphene films (also on TEM grids), conducting an extensive Raman analysis and providing thermal conductivity data. In the last section of the paper, the authors realized the transfer of MoS₂ films by the same approach. The authors also give their interpretation of the working mechanism of the cyclododecane-assisted method. Overall, this paper reports incremental results with respect to the present literature and, as such, does not possess the levels of originality, novelty and significance required to qualify for publication in *Nature Communications*. Personally, I would see this paper of potential interest for a journal like *Scientific Reports*.

That being said, before resubmitting to any journal the paper needs to be amended in its main, surprising flaw, which is the complete disregard for the previous literature and in particular for aforementioned papers that proposed and perfected the cyclododecane-assisted method.

In particular:

- 1) The introduction and the discussion of results should be rewritten considering past results.
- 2) The authors should analyze the literature on “clean transfer of graphene” at large areas (ideally wafer-scale) and highlight if and how their current results present sound advancements with respect to the state of the art.
- 3) The paper titles should be modified, mentioning the keyword “cyclododecane” in it.

This brings me to my last comment: I can hardly understand how and why the authors do not refer to the literature on the topic. The authors should explain and clarify the situation, selecting one of the two only possibilities below and replying to the questions:

1. Did the authors not know about the past literature on the cyclododecane-assisted method? If this is the case, this is a serious shortcoming, as it denotes a poor analysis of the literature when planning the experiments and writing the paper. I remark that a one-second search with “graphene cyclododecane” as entry on Google returns links to the aforementioned APL papers in the first 10 and more results.
OR
2. If the authors knew about the past literature on the cyclododecane-assisted method, why did they choose to omit any reference to the previous papers in the introduction and in the discussion of the results?

In closing, here below I report Fig. 3 from “Solvent-free transfer of monolayer graphene with recrystallized cyclododecane” and Figs 2 and 3 from the present submission.

[REDACTED]

Fig. 3 from “Solvent-free transfer of monolayer graphene with recrystallized cyclododecane”.

Figs 2 and 3 from “High-intactness and clean transfer for fabricating suspended two-dimensional Materials” (present submission).

Reviewer #3 (Remarks to the Author):

In this work, the authors introduce a novel method for transferring ultraclean suspended 2D materials using non-toxic, volatile cyclododecane (C₁₂H₂₄) as a support medium. This approach is particularly attractive to the 2D materials community, given the critical need for reliable and high-quality transfer processes for practical device applications. The manuscript is well-organized and clearly written. However, similar transfer techniques, especially those involving small or volatile molecules like paraffin, have been reported and are considered challenging to control precisely due to factors such as the volatility rate, which depends on various environmental conditions, complicating large-scale industrial applications. Additionally, as with many transfer methods, more evidence is needed to substantiate claims of cleanliness and completeness of the transfer, which tends to be limited to localized characterization, such as AFM and TEM, covering only micron and sub-micron scales. Recent studies have begun to employ electrical and spectroscopic characteristics for broader area partial mapping, providing more objective evidence of the uniformity and reliability of the transfer process, crucial for assessing the quality of transferred 2D materials. This aspect is almost absent from the current study, even though suspended 2D samples are presented, which could also corroborate the novelty and integrity of the transfer method for on-substrate samples.

This paper cannot be accepted in its current form, but it can be considered for publication until the following issues are addressed:

1. The introduction "However, without the support from the underlying substrates, free-standing graphene are prone to be teared by uncontrollable interfacial forces during the removal of the polymers." The statement highlights that high-integrity transfer of graphene is not solely dependent on the supporting layer but also significantly relies on the wet process, specifically during solvent removal. The surface tension force is widely recognized for causing the rupture of 2D materials. The authors can refer to previous studies, such as but not limited to Nanoscale, 2016, 8, 3555–3564, the selecting solvents with relatively lower surface energy can be used to avoid the drag force issue arising from solvent drying. Furthermore, thermal annealing is utilized to remove carbon contamination from the suspended graphene, which results from polymer residue. Although some residue may still remain on the surface, the authors can discuss these addressed issues and any progress made. This discussion could provide updated insights into this work.
2. The manuscript lacks quantitative descriptions of the adhesion energy of cyclododecane on graphene and its mechanical properties, such as Young's modulus, which could be evaluated through density functional theory (DFT). This information is vital to support claims of the robust mechanical properties of cyclododecane during wet etching and transferring processes.
3. On page 7, under "Fabrication of Other Free-Standing 2D Materials," the author demonstrates that typical 2D MoS₂ can still be successfully transferred onto a holey substrate. This transfer is based on mechanically exfoliated flakes. However, the typical process for transferring 2D materials typically involves CVD-grown 2D materials on sapphire, which are then decoupled in water or a base solution (NaOH, KOH). Can CD also be shown to transfer wafer-scale MoS₂ from sapphire to SiO₂ or a holey substrate?
4. The manuscript relies on localized morphological analysis (TEM, AFM) to argue for the cleanliness and integrity of the transfer process. However, electrical properties, more sensitive to the quality of 2D material transfers, could offer a broader and more accurate assessment.

Statistical analysis of electrical characteristics, such as mobility and FET features, across the transfer area (spatial mapping), would better demonstrate large-area uniformity.

5. On the annealing of sapphire wafers in pure oxygen at 1020°C for 6 hours: the impact of using pure oxygen on the prepared Cu on sapphire should be discussed, citing relevant studies if this effect has been reported.

6. The experimental description of the cyclododecane process lacks clarity, including missing information on chemical specifications, concentrations, conditions, coating methods, environmental pressure, sublimation pressure, heating methods, and duration.

7. Differences in wrinkle formation on graphene transferred onto different substrates require detailed explanation, especially considering the stress introduced during film transfer to substrates, which may result in rippling that maybe observed at higher magnification SEM.

8. The etching of Cu with Na₂S₂O₈ solution and the potential for metal residue require quantitative discussion, possibly using XPS for elemental analysis to ensure the removal of metal ions like Na and Cu, addressing the limitations of simple DI water cleaning.

9. The paper claims higher cleanliness and integrity of transferred materials but lacks quantitative descriptions, particularly regarding the proportion of residues. AFM could provide a more suitable characterization method for this analysis.

Reviewers' Comments:

(Comments in **black**, response in **blue** and revised text highlighted in **yellow**)

Reviewer #1 (Remarks to the Author):

In this paper, the authors demonstrate the use of cyclododecane as a transfer medium for the fabrication of suspended graphene and other 2D material membranes with high yield. AFM topography mapping, TEM and Raman mapping analysis has been used to demonstrate the cleanliness and low built-in strain and doping of the suspended graphene. Raman is also used to confirm the high thermal conductivity which can be linked to the cleanliness.

We thank the reviewer very much for making valuable comments on our work. All the comments are responded point by point.

1. CD was introduced as a sacrificial transfer medium for graphene 10 years ago, but this paper does not acknowledge the literature that exists, for instance, Appl. Phys. Lett. 105, 113101 (2014) and Brajpuriya, R. JMR 36 (2021). While this was not for transferring suspended membranes, nonetheless it represents important prior art for the use of CD for graphene transfer and must be noted explicitly.

Response: Thank you for your constructive suggestion. We sincerely apologize for omitting the mentioned references regarding the use of CD as a transfer medium. As suggested by the reviewers, we have now included the suggested papers as references [29] and [30], respectively. Additionally, we have cited other CD-related papers.

The revised part,

Here, we propose a universal transfer strategy to fabricate high-intactness and ultraclean suspended 2D materials by using non-toxic volatile cyclododecane (CD, C₁₂H₂₄) to support 2D materials. CD is firstly reported as a clean transferring medium by Capasso et al. in 2014²⁹ which can efficiently transfer CVD grown graphene to silicon wafer substrate³⁰⁻³⁴, or onto indium tin oxide (ITO) glass substrate for solar cells applications³⁵.

References:

[29] Capasso, A. *et al.* Cyclododecane as support material for clean and facile transfer of large-area few-layer graphene. *Applied Physics Letters* **105** (2014).

[30] Brajpuriya, R. Chemical vapor deposition of graphene by ethanol decomposition and its smooth transfer. *Journal of Materials Research* **36**, 3258-3266 (2021).

- [31] Kim, M. J. *et al.* Solvent-free transfer of monolayer graphene with recrystallized cyclododecane. *Applied Physics Letters* **123** (2023).
- [32] Capasso, A. *et al.* Nitrogen-doped graphene films from chemical vapor deposition of pyridine: influence of process parameters on the electrical and optical properties. *Beilstein Journal of Nanotechnology* **6**, 2028-2038 (2015).
- [33] Lisi, N. *et al.* Contamination-free graphene by chemical vapor deposition in quartz furnaces. *Scientific Reports* **7** (2017).
- [34] Gnisci, A. *et al.* Ethanol-CVD Growth of Sub-mm Single-Crystal Graphene on Flat Cu Surfaces. *The Journal of Physical Chemistry C* **122**, 28830-28838 (2018).
- [35] Capasso, A. *et al.* Chemical Vapor Deposited Graphene-Based Derivative As High-Performance Hole Transport Material for Organic Photovoltaics. *ACS Applied Materials & Interfaces* **8**, 23844-23853 (2016).

2. *The authors themselves have previously published the transfer of suspended graphene on TEM grids (without the use of CD), but these references (12, 13) are buried in the first generic sentence and not explicitly noted. This is particularly significant because the phenomenon described in fig 4 of this paper was already noted in Ref 12, but there is no explicit reference to this past work. This should be corrected.*

Response: Thank you for your valuable suggestion, and we greatly appreciate the reviewer's attention to our previous works on the fabrication of suspended graphene on TEM grid. In Fig. 4c, traditional single-layer graphene can be easily broken in water drying process for the surface tension of water, as analyzed in our previous work (Ref [12] in the revised manuscript). To reduce the crack resulted from surface tension, transfer-free strategy by selectively etching Cu substrate in Ref [12] and using CD transferring media in this work are effective ways. According to this kind mention, we have now included a citation to this paper at the corresponding section and provided a related discussion in the revised manuscript, please refer to the following revision.

The revised part,

It has been reported that the surface tension of water is measured to be approximately 72.8 mN/m at room temperature⁴⁵; Such a high surface tension of water would cause a remarkable stress of graphene (γ_{s-l}) especially for single-layer graphene whose stress is the largest comparing to few-layer graphene, and the catastrophic rupture can be expected once the fracture strength of graphene is reached¹².

References

[12] Zheng, L. et al. Robust ultraclean atomically thin membranes for atomic-resolution electron microscopy. Nat Commun 11, 541 (2020).

3. Noting the above omissions, this work combines the previous work from the authors on suspended graphene transfer with the use of CD which has been demonstrated for graphene transfer before, to fabricate high-yield large-diameter suspended graphene membranes. This does indeed represent a significant advancement in the state of the art and worthy of publication.

Response: We appreciate the reviewer very much for your comprehensive evaluation and high recognition of our work. We sincerely appreciate the time you have dedicated to reviewing our manuscript and providing constructive comments. Focusing on high-intactness and clean transfer for fabricating suspended 2D material membranes, it is expected that this work can attract interests from a broad range of readers and promote potential applications in related fields.

4. There are numerous language errors in this manuscript and the authors are encouraged to undertake professional language revision of this paper to ensure a high language standard prior to publication.

Response: Thanks for putting the language errors in this manuscript. We have taken a full language check for the whole manuscript and the revised parts are highlighted.

The revised part,

a high ratio of the intensity of 2D peak (I_{2D}) to the intensity of G peak (I_G) and no defect-related D peak of the suspended graphene transferred by CD confirm reduced doping and strain level.

Exposed to air, the sublimation was observed to be faster at the edge of TEM grids, which usually occurred from edge to the center. As illustrated in the corresponding OM images and the corresponding **Supplementary Video 2**, after 70 minutes, the sublimation of CD was complete, leaving free-standing graphene intact.

Reviewer #2 (Remarks to the Author):

1. The paper “*High-intactness and clean transfer for fabricating suspended two-dimensional materials*” from Wang et al. focuses on the cyclododecane-assisted method to transfer CVD graphene. The method has been proposed for few-layer graphene in 2014 by Capasso et al. and is well known in the 2D materials community (the original paper has more than 60 citations - *Cyclododecane as support material for clean and facile transfer of large-area few-layer graphene*, 10.1063/1.4895733). In 2023, the same group proposed a refinement of the cyclododecane-assisted method to afford large-area transfer of monolayer graphene (*Solvent-free transfer of monolayer graphene with recrystallized cyclododecane*, 10.1063/5.0169748)

(Incidentally, these are some more papers, among the others, utilizing the cyclododecane-assisted method:

Nitrogen-doped graphene films from chemical vapor deposition of pyridine: influence of process parameters on the electrical and optical properties, 10.3762/bjnano.6.206

Chemical vapor deposited graphene-based derivative as high-performance hole transport material for organic photovoltaics, 10.1021/acsami.6b06749

Contamination-free graphene by chemical vapor deposition in quartz furnaces
10.1038/s41598017-09811-z

Ethanol-CVD growth of sub-mm single-crystal graphene on flat cu surfaces,
10.1021/acs.jpcc.8b10094

In the submitted paper, Wang et al. used cyclododecane-assisted method to prepare suspended graphene films (also on TEM grids), conducting an extensive Raman analysis and providing thermal conductivity data. In the last section of the paper, the authors realized the transfer of MoS₂ films by the same approach. The authors also give their interpretation of the working mechanism of the cyclododecane-assisted method. Overall, this paper reports incremental results with respect to the present literature and, as such, does not possess the levels of originality, novelty and significance required to qualify for publication in Nature Communications. Personally, I would see this paper of potential interest for a journal like Scientific Reports.

That being said, before resubmitting to any journal the paper needs to be amended in its main, surprising flaw, which is the complete disregard for the previous literature and in particular for aforementioned papers that proposed and perfected the cyclododecane-assisted method.

In particular:

- 1) *The introduction and the discussion of results should be rewritten considering past results.*

Response: We appreciate the reviewer's comments on our work and we highly respect these opinions. Indeed, it is really true that there is an important research work which first reported CD medium to transfer few-layer graphene membranes onto flat Si/SiO₂ substrate in 2014. Despite the same transfer medium, our work focused on efficient and clean fabrication of suspended 2D materials, the involved transfer processes are significantly different from the reported way, which in turn enable the high-intactness fabrication of suspended graphene.

We believe the novelty and achievements in this work on the fabrication of large-area suspended graphene is sufficient, as reviewer 1 put that our work does indeed represent a significant advancement in the state of the art and worthy of publication. In detail, we would like to emphasize the breakthrough of the preparation of suspended graphene in this work, and explain the reason why this manuscript matches the scope of *Nature Communications*. Following response contents were carefully organized from three aspects, (a) The differences of CD based transfer details between this work and the reported works, (b) The achievements in this work, (c) Summary of the novelty.

(a) The differences between this work and the related reported works using CD transferring medium

a1. The goal substrate is different. In the previously reported works, the primary focus was on transferring graphene onto nonporous flat substrates like Si wafers and glass. In contrast to those works, this study aimed to achieve high-quality fabrication of suspended 2D materials. To accomplish this, we utilized CD as the support layer to transfer the 2D materials onto TEM holey substrates. This approach allowed for the creation of suspended 2D materials with improved quality and characteristics.

Different goal substrates

Figure R1. Illustration on different goal substrates in the reported works and this work. The left image is Figure 1b extracted from Ref [29], and the right image is the fabricated single-layer suspended graphene in Figure. 1e from this work.

a2. The way to form CD layer is different. In this work, CD (cyclic alkane) was utilized as a transfer medium for transferring graphene. The good adhesion between graphene and CD enables sufficient contact between CD on the surface of graphene, which is key for avoiding the crack formation of graphene, especially for suspended graphene. In reported works, CD was dissolved in organic solutions such as hexane, and CD layer was formed via spin coating approach. However, in our work, a different approach was employed. CD solid particles were directly melted at 70°C on substrate, allowing CD to fully cover the Cu/graphene substrate through surface wetting (Supplementary Video 1). And followed by cooling it to room temperature to form CD layer. Our method ensures that the full coverage of CD layer on rough graphene surface on Cu; in such case, no gap between graphene and CD would be formed, ensuring the crack-free transfer. By directly melting and utilizing CD as a transfer medium, we achieved a high-efficiency and clean suspended 2D material fabrication.

The revised part,

After attaching TEM grid onto graphene assisted by evaporation of isopropanol (IPA), CD particles around 1 mg were introduced on graphene surface. CD was then melted

through heating to fully cover graphene surface (**Supplementary Video 1**), and it returned to solid state to form a stable compact supporting layer at room temperature.

Figure R2. Illustration on the different ways to form CD layer in reported works and this work.

The way to form CD layer is different

Spin coating

Reported ways

extracted from Fig. 7 in Ref [30] [REDACTED]

CD is diluted in ethyl ether or cyclohexane solution, etc.

This work

Extracted from Fig 1a and Supplementary video 1

By melting CD solid particles, solvent-free

The left image is extracted from Ref [30], by diluting CD in organic solvent and spin coating to form CD layer. The right image shows the way by melting CD solid particles to form CD layer in this work.

(b) The achievements in this work

As for suspended 2D materials fabrication, achieving high-intactness, clean and mass fabrication simultaneously remains a significant challenge, particularly for monolayer 2D material structures. Here, we successfully addressed these challenges by utilizing volatile CD as a transfer medium, resulting in the fabrication of suspended graphene with 99% intactness and a clean surface. The solid property of CD provides support to ensure the high intactness of the graphene during the transfer process. Additionally, CD has the capability of complete removal by sublimation at room temperature, which guarantees a clean surface of suspended graphene. This is in contrast to traditional transferring methods that involve the use of polymers, which can leave contamination residues (*Nano Lett* 12, 414-419 (2012), *RSC Adv* 6, 83954-83962 (2016)), or liquid tension forces, which can cause breakage effects (*Adv Mater* 29 (2017), *Adv Mater* 34 (2022)). Furthermore, leveraging these advantages, mass fabrication of clean monolayer suspended graphene was achieved, demonstrating the huge potential of this strategy (Fig. 1g, in the main text).

By employing volatile CD as a transfer medium, we effectively addressed the challenges of intactness, cleanliness, and mass production in the fabrication of suspended 2D materials, showing the promising prospects of this approach.

(c) Summary of the novelty in this work

The fabrication of high-quality monolayer suspended graphene and other suspended 2D materials is of great significance in exploring their intrinsic properties and high-resolution TEM characterization. The main achievements of this work include the proposal and realization of high-intactness and clean monolayer suspended graphene. Additionally, we demonstrated the fabrication of suspended MoS₂ from mechanically peeled and CVD-grown MoS₂. Considering the impact and research focus of *Nature Communications*, which is a high impact journal covering a wide range of areas, this work aligns well with the journal's scope and represents a breakthrough in the **high-quality fabrication of suspended 2D materials**.

Finally, we sincerely apologize that we had not cited the important work on using CD to transfer graphene onto flat substrate. We thank the reviewer very much to give us valuable comments to revise the manuscript, we have cited and discussed these related literatures in our revised manuscript. Once again, we sincerely thank the reviewer for pointing this out.

The revised part, as we responded to reviewer 1 above,

Here, we propose a universal transfer strategy to fabricate high-intactness and ultraclean suspended 2D materials by using non-toxic volatile cyclododecane (CD, C₁₂H₂₄) to support 2D materials. CD is firstly reported as a clean transferring medium by Capasso et al. in 2014²⁹ which can efficiently transfer CVD grown graphene to silicon wafer substrate³⁰⁻³⁴, or onto indium tin oxide (ITO) glass substrate for solar cells applications³⁵.

References:

- [29] Capasso, A. *et al.* Cyclododecane as support material for clean and facile transfer of large-area few-layer graphene. *Applied Physics Letters* **105** (2014).
- [30] Brajpuriya, R. Chemical vapor deposition of graphene by ethanol decomposition and its smooth transfer. *Journal of Materials Research* **36**, 3258-3266 (2021).
- [31] Kim, M. J. *et al.* Solvent-free transfer of monolayer graphene with recrystallized

cyclododecane. *Applied Physics Letters* **123** (2023).

[32] Capasso, A. *et al.* Nitrogen-doped graphene films from chemical vapor deposition of pyridine: influence of process parameters on the electrical and optical properties. *Beilstein Journal of Nanotechnology* **6**, 2028-2038 (2015).

[33] Lisi, N. *et al.* Contamination-free graphene by chemical vapor deposition in quartz furnaces. *Scientific Reports* **7** (2017).

[34] Gnisci, A. *et al.* Ethanol-CVD Growth of Sub-mm Single-Crystal Graphene on Flat Cu Surfaces. *The Journal of Physical Chemistry C* **122**, 28830-28838 (2018).

[35] Capasso, A. *et al.* Chemical Vapor Deposited Graphene-Based Derivative As High-Performance Hole Transport Material for Organic Photovoltaics. *ACS Applied Materials & Interfaces* **8**, 23844-23853 (2016).

- 2) *The authors should analyze the literature on “clean transfer of graphene” at large areas (ideally wafer-scale) and highlight if and how their current results present sound advancements with respect to the state of the art.*

Response: Thank the reviewer for raising this comment. We want to make an explanation that **this work primarily concentrates on the high-quality fabrication of suspended 2D materials rather than transferring large-area monolayer graphene onto a flat substrate.** In this work, the transfer of graphene onto a flat silicon wafer was performed solely to demonstrate the cleanliness of the CD-based transfer method using AFM. This was necessary because AFM characterization of suspended graphene may not accurately reflect the surface morphology, when graphene is suspended (as depicted in Figure 2d).

Transferring graphene onto a flat substrate is indeed a hot research topic, especially for achieving high-quality transfers onto large areas, such as 4-inch wafers. However, using CD as transferring medium, it is hard to achieve large area transfer of monolayer graphene due to the properties of CD. By using the melting-based method, CD has the ability to duplicate the surface morphology of the growth substrate. However, achieving good conformal contact between monolayer graphene and a flat Si wafer over a large area becomes difficult due to the high rigidity of CD. We have emphasized this point clearly in the revised version of manuscript.

In fabricating suspended 2D materials, particularly single-layer suspended 2D materials, it is quite a large area for several tens of micrometers. In our transfer process,

CD can effectively bond 2D thin films with flexible TEM grids through the holes on the grid. This bonding ensures good contact between the 2D thin films and the TEM substrate, resulting in the fabrication of high-quality suspended 2D materials. Therefore, CD is more suitable for transferring 2D materials to flexible holey substrates compared to flat rigid substrates, over large area (wafer scale). In our work, we successfully achieved full coverage of intact single-layer suspended graphene with a diameter exceeding 36 μm .

Furthermore, our research focuses on high-quality fabrications of suspended 2D materials, and we have analyzed and organized the current research results in the Introduction section from two perspectives: strategies for achieving high-intactness and strategies for achieving clean fabrications. We believe that our research topic and analytical approach are clear and well-defined.

The revised part,

At introduction part,

In this regard, the intactness of suspended area can be improved by replacing acetone with low surface tension solvent such as methoxy-nonafluorobutane ($\text{C}_4\text{F}_9\text{OH}_3$) after the removal of PMMA²¹⁻²³.

In traditional fabrication, the removal of supporting polymer is usually incomplete, thermal annealing at high temperature^{24,25} and plasma treatment²⁶ are further utilized to remove residual polymer. However, severe conditions sometimes cause amorphous carbon and defects in graphene^{27,28}.

At the section about clean transfer with CD,

In addition, one important thing should be noted that it is hard to achieve good contact between silicon wafer substrate and 2D films because CD is quite rigid. While in fabricating suspended 2D thin films on TEM holey grid, CD can bond 2D thin films with TEM grid well together through the holes on TEM grid. This bonding ensures the good contact between 2D thin films and TEM substrate to achieve high yield and high-intactness suspended 2D materials.

References:

[21] Chen, Y. M. *et al.* Ultra-large suspended graphene as a highly elastic membrane for capacitive pressure sensors. *Nanoscale* **8**, 3555-3564 (2016).

[22] Wang, Q. *et al.* Large-Size Suspended Mono-Layer Graphene Film Transfer Based on the

- Inverted Floating Method. *Micromachines (Basel)* **12** (2021).
- [23] Lee, C.-K. *et al.* Monatomic Chemical-Vapor-Deposited Graphene Membranes Bridge a Half-Millimeter-Scale Gap. *Acs Nano* **8**, 2336-2344 (2014).
- [24] Yulaev, A. *et al.* Toward Clean Suspended CVD Graphene. *RSC Adv* **6**, 83954-83962 (2016).
- [25] Lin, Y. C. *et al.* Graphene annealing: how clean can it be? *Nano Lett* **12**, 414-419 (2012).
- [26] Ferrah, D. *et al.* XPS investigations of graphene surface cleaning using H₂- and Cl₂-based inductively coupled plasma. *Surface and Interface Analysis* **48**, 451-455 (2016).
- [27] Zhuang, B., Li, S., Li, S. & Yin, J. Ways to eliminate PMMA residues on graphene -superclean graphene. *Carbon* **173**, 609-636 (2021).
- [28] Wang, X. *et al.* Direct Observation of Poly(Methyl Methacrylate) Removal from a Graphene Surface. *Chemistry of Materials* **29**, 2033-2039 (2017).

3) *The paper titles should be modified, mentioning the keyword “cyclododecane” in it.*

Response: Thank the reviewer for raising this valuable suggestion. We have added “cyclododecane” in title, which is modified as “**Cyclododecane based** high-intactness and clean transfer for fabricating suspended two-dimensional materials” in the revised manuscript.

This brings me to my last comment: I can hardly understand how and why the authors do not refer to the literature on the topic. The authors should explain and clarify the situation, selecting one of the two only possibilities below and replying to the questions:

- 1. Did the authors not know about the past literature on the cyclododecane-assisted method? If this is the case, this is a serious shortcoming, as it denotes a poor analysis of the literature when planning the experiments and writing the paper. I remark that a one-second search with “graphene cyclododecane” as entry on Google returns links to the aforementioned APL papers in the first 10 and more results. **OR***
- 2. If the authors knew about the past literature on the cyclododecane-assisted method, why did they choose to omit any reference to the previous papers in the introduction and in the discussion of the results?*

*In closing, here below I report Fig. 3 from “**Solvent-free transfer of monolayer graphene with recrystallized cyclododecane**” and Figs 2 and 3 from the present submission.*

[REDACTED]

Fig. 3 from “Solvent-free transfer of monolayer graphene with recrystallized cyclododecane”.

Figs 2 and 3 from “High-intactness and clean transfer for fabricating suspended two-dimensional Materials” (present submission).

Response: We are sorry for omitting the discussion on the mentioned cyclododecane-assisted transfer methods. As we responded above, there are obvious differences between this work and the previously reported cyclododecane-assisted works, as shown in the Figures listed by the reviewer. The reported works primarily focused on supported graphene transfer, whereas our work specifically emphasized and focused on the fabrication of suspended graphene. In Figure 2a-c in our manuscript, the AFM

measurements on supported graphene were included only to showcase the cleanness achieved through the CD-based transfer method. In our work, the other measurements are mainly based on suspended 2D materials. We respect the achievements in the reported cyclododecane-assisted works and we hope the reviewer could recognize the different achievements of our work in comparison with previous work. According to the reviewer's concern, we have included the mentioned references in the introduction section.

Reviewer #3 (Remarks to the Author):

In this work, the authors introduce a novel method for transferring ultraclean suspended 2D materials using non-toxic, volatile cyclododecane (C₁₂H₂₄) as a support medium. This approach is particularly attractive to the 2D materials community, given the critical need for reliable and high-quality transfer processes for practical device applications. The manuscript is well-organized and clearly written. However, similar transfer techniques, especially those involving small or volatile molecules like paraffin, have been reported and are considered challenging to control precisely due to factors such as the volatility rate, which depends on various environmental conditions, complicating large-scale industrial applications. Additionally, as with many transfer methods, more evidence is needed to substantiate claims of cleanliness and completeness of the transfer, which tends to be limited to localized characterization, such as AFM and TEM, covering only micron and sub-micron scales. Recent studies have begun to employ electrical and spectroscopic characteristics for broader area partial mapping, providing more objective evidence of the uniformity and reliability of the transfer process, crucial for assessing the quality of transferred 2D materials. This aspect is almost absent from the current study, even though suspended 2D samples are presented, which could also corroborate the novelty and integrity of the transfer method for on-substrate samples.

This paper cannot be accepted in its current form, but it can be considered for publication until the following issues are addressed:

We are very grateful for your constructive evaluation on our work. Based on these valuable comments, we have made a careful correction with the comments responded point by point.

1. The introduction "However, without the support from the underlying substrates, free-

standing graphene are prone to be teared by uncontrollable interfacial forces during the removal of the polymers.” The statement highlights that high-integrity transfer of graphene is not solely dependent on the supporting layer but also significantly relies on the wet process, specifically during solvent removal. The surface tension force is widely recognized for causing the rupture of 2D materials. The authors can refer to previous studies, such as but not limited to *Nanoscale*, 2016, 8, 3555–3564, the selecting solvents with relatively lower surface energy can be used to avoid the drag force issue arising from solvent drying. Furthermore, thermal annealing is utilized to remove carbon contamination from the suspended graphene, which results from polymer residue. Although some residue may still remain on the surface, the authors can discuss these addressed issues and any progress made. This discussion could provide updated insights into this work.

Response: Thanks for the reviewer’s constructive suggestion. It's indeed important to consider research progress in using solvents with low surface energy to minimize damage and employing thermal annealing to reduce contamination when transferring 2D materials with supporting polymer. According to this suggestion, we have added these related contents in revised manuscript at the introduction part.

(a) using solvents with low surface energy to reduce damage

Without the support from the underlying substrates, free-standing graphene films are prone to be teared by uncontrollable interfacial forces during the wet process. In this regard, the intactness of suspended area can be improved by replacing acetone with low surface tension solvent such as methoxynonaflurorbutane ($C_4F_9OH_3$) after the removal of PMMA²¹⁻²³.

(b) using thermal annealing to reduce contamination

In traditional fabrication, the removal of supporting polymer is usually incomplete, thermal annealing at high temperature^{24,25} and plasma treatment²⁶ are further utilized to remove residual polymer. However, severe conditions sometimes cause amorphous carbon and defects in graphene^{27,28}.

References

- [21] Chen, Y. M. *et al.* Ultra-large suspended graphene as a highly elastic membrane for capacitive pressure sensors. *Nanoscale* **8**, 3555-3564 (2016).
- [22] Wang, Q. *et al.* Large-Size Suspended Mono-Layer Graphene Film Transfer Based on the Inverted Floating Method. *Micromachines (Basel)* **12** (2021).

- [23] Lee, C.-K. *et al.* Monatomic Chemical-Vapor-Deposited Graphene Membranes Bridge a Half-Millimeter-Scale Gap. *Acs Nano* **8**, 2336-2344 (2014).
- [24] Yulaev, A. *et al.* Toward Clean Suspended CVD Graphene. *RSC Adv* **6**, 83954-83962 (2016).
- [25] Lin, Y. C. *et al.* Graphene annealing: how clean can it be? *Nano Lett* **12**, 414-419 (2012).
- [26] Ferrah, D. *et al.* XPS investigations of graphene surface cleaning using H₂- and Cl₂-based inductively coupled plasma. *Surface and Interface Analysis* **48**, 451-455 (2016).
- [27] Zhuang, B., Li, S., Li, S. & Yin, J. Ways to eliminate PMMA residues on graphene -superclean graphene. *Carbon* **173**, 609-636 (2021).
- [28] Wang, X. *et al.* Direct Observation of Poly(Methyl Methacrylate) Removal from a Graphene Surface. *Chemistry of Materials* **29**, 2033-2039 (2017).

2. *The manuscript lacks quantitative descriptions of the adhesion energy of cyclododecane on graphene and its mechanical properties, such as Young's modulus, which could be evaluated through density functional theory (DFT). This information is vital to support claims of the robust mechanical properties of cyclododecane during wet etching and transferring processes.*

Response: Thanks for the reviewer's good suggestion. It is a truly good idea and we highly agree the way to reflect the adhesion energy of cyclododecane on graphene through theoretical calculation. Here, we sincerely apologize that we have not finished the theoretical calculation on Young's modulus. Instead, we carried mechanical test of CD, and its Young's modulus is calculated as 102.8 Mpa (Supplementary Fig. 17a). Besides, we presented the mechanical property of CD through the strong support it provides to TEM grids, as illustrated in Figure 1b, Supplementary Fig. 1, and Supplementary Fig. 17b. The CD solid compact layer offers robust support to graphene, which plays a crucial role in protecting the 2D materials throughout the entire transfer process. Particularly, it helps prevent cracks caused by surface tension forces when lifting 2D materials from the liquid solution during etching and washing processes.

The revised part,

From the mechanical compression curve of CD, the corresponding Young's modulus is calculated to be 102.8 Mpa, such strong mechanical properties of CD ensure sufficient support from compact CD layer to graphene for large crack-free suspended area (Supplementary Fig. 17).

Fig. 1(b) The optical image of the as-transferred graphene on TEM grid with the assistance of CD.

Supplementary Fig. 1(b) The image of CD/TEM grid/graphene after completely etching Cu substrate, the material around TEM grid and in light white color is CD.

Supplementary Fig. 17. Mechanical properties of CD. (a) Mechanical compression curve for CD. **(b)** Demonstration for CD to support objects.

Mechanical comprehension curve is tested by the electronic universal testing machine (Sinotest Equipment Co. Ltd., DDL1003050301). The measured CD sample is in size of 14 mm diameter and 21.5 mm height. Young's modulus E is calculated from the comprehension curve of CD sample at the elastic region (linear region) by the following equation:

$$E = \frac{FL}{S\Delta L} = \frac{\sigma}{\varepsilon}$$

In which, F is the applied force, L is the height, S is the cross-sectional area, ΔL denotes the length variation, $\sigma = F/S$ represents the stress and $\varepsilon = \Delta L/L$ represents the strain.

3. On page 7, under "Fabrication of Other Free-Standing 2D Materials," the author demonstrates that typical 2D MoS₂ can still be successfully transferred onto a holey substrate. This transfer is based on mechanically exfoliated flakes. However, the typical process for transferring 2D materials typically involves CVD-grown 2D materials on sapphire, which are then decoupled in water or a base solution (NaOH, KOH). Can CD also be shown to transfer wafer-scale MoS₂ from sapphire to SiO₂ or a holey substrate?

Response: Thanks for the reviewer's good suggestion. According to this suggestion and considering the advantage of CD in fabricating suspended 2D materials, we have conducted the experiment about transferring CVD-grown MoS₂ from sapphire substrate to holey TEM grid. To achieve this transfer, PMMA and TRT are used, which are attached before coating PVA. The purpose of this process is to facilitate the peeling off of MoS₂ from the sapphire substrate in NaOH solution. By utilizing this approach, we have successfully transferred CVD-grown MoS₂ onto the desired holey TEM grid.

The revised part,

Different from transferring mechanically exfoliated MoS₂, transferring CVD-grown MoS₂ to fabricate suspended MoS₂ films requires polymethyl methacrylate (PMMA) and thermal release tape (TRT) to peel off MoS₂ from sapphire grown substrate in NaOH solution before coating PVA layer (see **Supplementary Fig. 19** and Method section for details).

Supplementary Fig. 19. Fabrication procedures for suspended MoS₂ films by transferring CVD-grown MoS₂.

And the detailed transferring steps in Experimental Section,

Fabrication procedures for suspended MoS₂ films by transferring CVD-grown MoS₂.

The transferred MoS₂ thin film was grown on sapphire substrate. First, polymethyl methacrylate (PMMA, average Mw ≈ 910 000, Macklin) was dissolved in anisole with 4 wt% under magnetic stirring at 60 °C for several hours. This solution was

subsequently spun onto CVD-grown MoS₂ at 1000 rpm for 60 s, repeated twice. Afterward, the sample was heated at 60°C for 5 minutes to form a PMMA thin layer. Then, thermal release tape (TRT, 3198MS, Nitto Denko) was attached onto PMMA layer. After immersing the whole stack in NaOH solution (1M) for 5 min, MoS₂ film was delaminated from sapphire. Thereafter, the prepared MoS₂/PMMA/TRT was washed in deionized water for several times. After the sample became dry, 10 wt% PVA was spin coated on MoS₂ layer at 1000 rpm for 60 s, repeated twice. And the sample was then heated at 60°C for 5 min. After that, polyethylene terephthalate (PET) layer was attached on PVA layer. Then, MoS₂ layer was directly peeled off from PMMA layer. Holey TEM grid was then attached on MoS₂/PVA/PET, and a little IPA solution was dropped to enhance the interaction between TEM grid and MoS₂. Subsequently, solid CD particles (≈ 1 mg) was introduced onto TEM grid by tweezers. After CD melted at 70°C, the whole stack was fully immersed in deionized water to remove PVA. Finally, after the complete CD sublimation at 40~50°C under air pressure, suspended MoS₂ on TEM grid was successfully fabricated.

We also carried SEM and Raman characterizations. Related contents are added in revised manuscript and Supplementary Information.

The revised part,

After transferring, the SEM image of the suspended MoS₂ flakes on TEM holey grid in **Fig. 5c** displays the same morphology with **Fig. 5b**, illustrating the high-intactness of the transferred MoS₂. Besides, transferring from sapphire grown substrate, SEM image of the fabricated suspended MoS₂ in **Supplementary Fig. 20** also proves the high-intactness transfer with the assistance of CD.

Supplementary Fig. 20. SEM images for suspended MoS₂ on TEM lacy grid transferring from CVD-grown MoS₂. (a) and (b) are the SEM images of suspended MoS₂ in different magnifications.

Thickness-dependent Raman spectrums with different layer numbers transferring from exfoliated MoS₂ flakes and for single-layer suspended MoS₂ transferring from sapphire substrate are presented in **Supplementary Fig. 21 and 22**, respectively.

Supplementary Fig. 22. Raman characterization for suspended MoS₂ on TEM grid transferring from CVD-grown MoS₂ films. The Raman peak frequency separation illustrates the monolayer feature of the transferred MoS₂.

4. The manuscript relies on localized morphological analysis (TEM, AFM) to argue for the cleanliness and integrity of the transfer process. However, electrical properties, more sensitive to the quality of 2D material transfers, could offer a broader and more accurate assessment. Statistical analysis of electrical characteristics, such as mobility and FET features, across the transfer area (spatial mapping), would better demonstrate large-area uniformity.

Response: Thanks for the reviewer's valuable comment. It is really true and we fully agree to reflect the quality of transferring by using analysis of electrical characteristics. We want to make an explanation that CD, being a small molecular substance, can easily crystallize at room temperature, making it quite rigid. **When using CD as the medium to transfer graphene onto a flat Si wafer, it is a challenge to achieve conformal contact between the graphene film and the large area of the flat Si wafer.** Therefore, it becomes difficult to obtain large area monolayer graphene/SiO₂/Si sample for characterizing the carrier mobility of transferred graphene. CD is more suitable in transferring 2D materials onto holey substrate rather than flat substrate. **The major highlight of this work lies in the fabrication of high-quality suspended 2D materials.** However, constructing a suspended graphene device for mobility testing, especially a suspended device array, is quite challenging due to the susceptibility of crack formation during the device fabrication, such as lift off and low-pressure evaporation. Alternatively, Raman FWHM_{2D} was used to reflect the carrier mobility. And we use this method to characterize the large-area uniformity of prepared graphene. Raman mapping result for 110×110 μm² intact suspended area in Supplementary Fig. 15 with a low value around 22 cm⁻¹ further reflects the high uniformity and improved electronic properties of suspended graphene across large-area obtained in this work.

The revised part,

Full width at half-maximum of 2D peak (FWHM_{2D}) is also reported to be an indicator of strain and doping³⁶, and can link to the carrier mobility of device³⁷. As presented in **Fig. 3c**, the statistical FWHM_{2D} of suspended graphene films in different TEM grids exhibits a low value around 22 cm⁻¹, confirming the improved quality owing to the clean surface free of additional doping and strains. In addition, the low FWHM_{2D} also determines high carrier mobility. Raman mapping result for 110×110 μm² intact suspended area in **Supplementary Fig. 15** further reflects the high large-area uniformity of electronic quality of suspended graphene obtained in this work.

Reference

[37] Robinson, J. A. *et al.* Correlating Raman Spectral Signatures with Carrier Mobility in Epitaxial Graphene: A Guide to Achieving High Mobility on the Wafer Scale. *Nano Letters* **9**, 2873-2876 (2009).

Supplementary Fig. 15. Raman characterization for suspended single-layer graphene. (a) The SEM image of suspended single-layer graphene on TEM SiN graphene with 5 μm diameter for each hole, the selected area is a 11×11 array with full intact graphene films. **(b)** Raman mapping and **(c)** statistical result of half-maximum of 2D peak ($\text{FWHM}_{2\text{D}}$) of suspended graphene at the selected area in **(a)**.

5. On the annealing of sapphire wafers in pure oxygen at 1020°C for 6 hours: the impact of using pure oxygen on the prepared Cu on sapphire should be discussed, citing relevant studies if this effect has been reported.

Response: We are thankful for the reviewer's comment. We have cited the related reference [54] (*ACS Nano* 2017, 11, 12, 12337, doi:10.1021/acsnano.7b06196) in our manuscript. The reference discussed the synthesis of single-crystal Cu (111) film on oxygen-pretreated sapphire wafers. The pre-annealing treatment of sapphire wafer in pure oxygen atmosphere facilitates the formation of more oxygen-terminated sapphire surface, contributing to the formation of intermediate ultrathin Cu_2O buffer layer. And

Cu₂O buffer layer can significantly alleviate the epitaxial stress between Cu and sapphire, thus motivating the epitaxy of the single-crystal Cu (111) film. In contrast, without pre-annealed in an oxygen atmosphere, the deposition of a Cu thin film on the sapphire followed by annealing in reducing atmosphere would produce in-plane twin boundaries with deep thermal grooves. Therefore, the pre-annealing treatment in an oxygen atmosphere plays a crucial role in attaining the desired single-crystal Cu (111) wafers.

According to the reviewer's concern, we have cited the related reference and included above discussion in the method section.

The revised part,

Before deposition, sapphire wafer was annealed in a pure oxygen atmosphere at 1020°C for 6 h under 1 atm pressure to ensure the quality of single-crystal Cu⁵⁴.

References

[54] Deng, B. *et al.* Wrinkle-Free Single-Crystal Graphene Wafer Grown on Strain-Engineered Substrates. *ACS Nano* **11**, 12337-12345 (2017).

6. *The experimental description of the cyclododecane process lacks clarity, including missing information on chemical specifications, concentrations, conditions, coating methods, environmental pressure, sublimation pressure, heating methods, and duration.*

Response: Thanks for the reviewer's constructive comment. Cyclododecane (CD) was purchased from Macklin company and used without any additional treatment. Different from traditional transferring, organic solvent is not needed to form CD layer *via* its liquid solution dissolved in some solvent at a certain concentration. In our transfer approach, solid CD particles with suitable amounts were directly placed on the target substance using tweezers. Then, heating at 70°C on a hot plate was applied to melt the CD and form a thin CD layer at room temperature. Alternatively, CD layer can also be achieved by spinning absolute CD melted solution at 70°C with 500 rpm speed for 10 s.

For the removal of CD, although CD can sublimate at room temperature, a higher temperature but below its melting point can speed up the sublimation process and reduce the total time. In this regard, the sublimation was conducted at 40°C under air pressure for approximately 6 hours to ensure complete removal of CD. According to

this comment, we have added related detailed information in the revised manuscript. Additionally, the whole process of CD, from its solid state to its melted liquid state under heating, covering the full surface of graphene, was displayed in Supplementary Video S1.

Below contents are the related revised parts,

After attaching TEM grid onto graphene assisted by evaporation of isopropanol (IPA), CD particles around 1 mg were introduced on graphene surface. CD was then melted through heating to fully cover graphene surface (**Supplementary Video 1**), and it returned to solid state to form a stable supporting layer at room temperature.

The removal of CD can be achieved by its spontaneous sublimation at room temperature. In the transferring, it is about 6 h to completely remove CD at 40°C under air pressure.

Transferring graphene procedures to Si wafer.

then solid CD particles (purchased from Macklin, ≈ 2 mg) were introduced onto the selected area of graphene film using clean tweezers. Next, the CD was melted at 70°C on a hot plate for several minutes, allowing CD to flow and fully cover the surface of the graphene film. Afterward, the sample was removed from the hot plate and allowed to cool down to room temperature, causing CD to return to its solid state and firmly attach to the graphene surface. Cu substrate was then etched with 0.5 M $\text{Na}_2\text{S}_2\text{O}_8$ solution, followed by washing with deionized water for several times. The CD/graphene was then taken out of water and placed to clean Si wafer. Finally, CD was completely removed by sublimating at 40°C under air pressure for roughly 6 hours or longer, and graphene on Si wafer was obtained.

The corresponding parts in fabricating suspended graphene films and MoS_2 films have also been supplemented. We wish the added detailed information could make the transferring methods clearer.

7. Differences in wrinkle formation on graphene transferred onto different substrates require detailed explanation, especially considering the stress introduced during film transfer to substrates, which may result in rippling that maybe observed at higher magnification SEM.

Response: Thanks for the reviewer's detailed suggestion. In this paper, we focus on high-intactness and clean fabrication methods of suspended 2D materials. As for

wrinkle, it can be usually observed in the transferred graphene. In our work, there are clear wrinkles in the fabricated suspended monolayer graphene (Fig. 1f). To the best knowledge of what we have known, the wrinkles on transferred graphene are not only generated in the high-temperature growth process but also in the transfer process (*J. Am. Chem. Soc.* 2011, 133, 17578–17581, 10.1021/ja207517u). Generally, more wrinkles can be produced when transferring graphene films from rough Cu foil compared with graphene on Cu/sapphire because corrugated surface existed in Cu foil. While in transferring especially in transferring graphene to rigid flat substrate, interface contact between graphene films and the target substrate is the main factor that determines the wrinkle formation. In traditional fabrication of supported graphene, using polymer with good flexibility and wettability as transferring medium (*Adv. Mater.* 2022, 34, 2105851, 10.1002/adma.202105851) is conducted to ensure the fine contact between graphene and substrates which, in turn, would reduce the density of wrinkles after the removal of transfer medium. Conversely, the choice of the transfer medium with poor wettability on the corrugated surface of graphene is proven to be the key for the formation of wrinkles. What's more, with soft target substrate, applying the tension and compression loading (*Nanoscale*, 2017, 9, 18180; *ACS nano* 3917 – 3925, 2015) or substrate engineering (*Nano Lett.* 2016, 16, 7121; *ACS Nano* 2020, 14, 166–174) can also be employed to fabricate the graphene wrinkle arrays.

With CD as transfer medium, this work displays obvious advantage in fabricating suspended graphene. With good wettability of CD on graphene, CD is able to bond graphene films with TEM grids well together through the holes on TEM grids before Cu etching, ensuring good conformal contact between graphene films and flexibility TEM holey grids. In our opinion, in the transferring, the good contact and the facile removal of CD by the sublimation would suppress the formation of wrinkles on the fabricated suspended graphene. The wrinkles on the suspended graphene are mostly produced in the growth process, which has little relationship with the kinds of holey substrates. To verify this, a lot of efforts will be devoted in the future to study in-situ AFM and SEM characterizations on the wrinkles of graphene films on Cu substrate before transferring and on TEM holey substrate after transferring.

8. The etching of Cu with Na₂S₂O₈ solution and the potential for metal residue require quantitative discussion, possibly using XPS for elemental analysis to ensure the

removal of metal ions like Na and Cu, addressing the limitations of simple DI water cleaning.

Response: Thanks for the reviewer's good suggestion. Accordingly, we have further carried XPS characterization. Both pure Si wafer and transferred graphene on Si wafer are analyzed, the results are given in Supplementary Fig. 11. From the results, graphene on Si wafer displayed stronger signal of C 1s peak compared with pure Si wafer. In addition, neither Na 1s and Cu 2p was observed, illustrating the quite clean surface washing by DI water to remove mental elements.

The manuscript is revised accordingly,

Firstly, after the sublimation of CD, optical microscopy (OM) image of graphene membrane transferred to Si/SiO₂ substrate exhibits a uniform contrast (**Fig. 2a**), and X-ray photoelectron spectroscopy (XPS) spectrums in **Supplementary Fig. 11** indicate no observable Na and Cu elements on graphene.

Supplementary Fig. 11 X-ray photoelectron spectroscopy (XPS) characterization. (a) XPS survey spectrums for pure Si wafer and the transferred graphene on Si wafer. **(b-c)** Enlarged XPS spectrums to detect Na and Cu elements, respectively.

9. The paper claims higher cleanliness and integrity of transferred materials but lacks quantitative descriptions, particularly regarding the proportion of residues. AFM could provide a more suitable characterization method for this analysis.

Response: Thanks for the reviewer's good suggestion. To quantitatively characterize the impurity residues in our transferred graphene with CD medium, we further carried systematical AFM characterization at different areas of the transferred graphene films onto SiO₂/Si wafer. The result is shown in Supplementary Fig. 11. By calculating the size ratio of impurity residues to graphene films in each AFM result, we get the statistic impurity residue ratio, and it is about 1.2%, which is shown in Supplementary Fig. 12.

The statistic illustrates a quite high-cleanness transferring with CD in this work.

The revised part,

Besides, the AFM images with little contaminations and a very low roughness also confirm the clean surface after the transfer (Fig. 2b, c). From some typical AFM images (Supplementary Fig. 12), the impurity residue is around 1.2% by calculating the area ratio of impurity-occupied region to the entire graphene films (Supplementary Fig. 13).

Supplementary Fig. 12 Atomic force microscopy (AFM) characterization of graphene films on SiO₂/Si wafer transferred by CD. (a-i) AFM images of graphene films at different areas.

Supplementary Fig. 13 Statistic on impurity residue of graphene films on SiO₂/Si wafer transferred by CD. The ratio is calculated by the ratio of observed impurity area to the characterized graphene films area through AFM image.

REVIEWER COMMENTS

Reviewer #3 (Remarks to the Author):

The author has diligently responded to most of the questions. Here is my following comments: The author mentions that "However, achieving good conformal contact between monolayer graphene and a flat Si wafer over a large area becomes difficult due to the high rigidity of CD. We have emphasized. This method is more suitable for small-scale suspended 2D transfer." I agree that the author has modified a previously known technique and found better transfer results on suspended or holey substrates (in terms of integrity and cleanliness) than before. In other words, I support the novelty in this part. However, the crux of the matter is that such a method indeed limits its widespread applicability. Since the author claims that this technique is mainly intended for high-quality suspended 2D, and explains its potential uses in basic 2D material properties exploration and for TEM purposes, the lack of demonstration of these target applications in the article to persuade readers that this improvement has significant benefits in these applications is a shortcoming.

Furthermore, regarding wrinkle formation, I acknowledge that the author has mentioned various complex factors, but I am curious to know whether this transfer method has indeed resulted in uniform material properties, especially to convince readers of the method's reliability. If the author believes SEM is more reflective of the morphology of suspended graphene, could an array sampling approach be used to highlight that these defects exist but are at least uniform over the whole suspended region? If the goal is to use it for fundamental 2D material property research or practical TEM cell applications, these wrinkle structures will inevitably cause an impact. Without this discussion, the current findings can only suggest that this is a method for high-yield transfer of 2D materials onto suspended substrates. Other questions are as follows:

Q2: The carrier film CD seems to be rigid, but when it is attached to the TEM grid, how is conformal contact between the graphene and the grid CD? If the removal of the CD is based on sublimation without a liquid phase (as shown in the supporting video), it seems difficult to maintain perfect interfacial contact. As depicted in Supporting Figure 1b, there are stresses present during the etching of Cu in a liquid phase and during the transfer process, which prevent tight adherence of the graphene to the TEM grid. Can the mechanism of directly sublimating the CD to remove it adequately explain this?

Q3: In Figure S20, the CVD-grown MoS₂ appears to be a non-continuous film, showing polycrystalline triangular domains. If so, the additional description can be provided to clarify this observation.

Q4: I understand that measuring mobility using the current structure might pose challenges for conducting FET or Hall measurements (though not impossible—it merely requires custom substrate fabrication and pre-layout of four electrodes before transfer). In the supplied Figure S15, a 2D mapping is shown where the width of the spectral features varies between 23.5 to 24 cm⁻¹ across different areas. How can one argue that this variation is uniform? Additionally, the cited literature uses this data to correlate with mobility. Would converting this mapping into a mobility distribution provide more compelling evidence of low defects and maintained uniformity across a

large area?

In Figure 13, how is the ratio of residue calculated, and is this residue from the CD or other sources? It needs to be defined first.

Moreover, could the authors specify the maximum size of the suspended region achievable with this method? At what size does the yield begin to drop below 80% (or even less)?

Reviewer #3 (Remarks to the Author):

The author has diligently responded to most of the questions. Here is my following comments:

The author mentions that "However, achieving good conformal contact between monolayer graphene and a flat Si wafer over a large area becomes difficult due to the high rigidity of CD. We have emphasized. This method is more suitable for small-scale suspended 2D transfer." I agree that the author has modified a previously known technique and found better transfer results on suspended or holey substrates (in terms of integrity and cleanliness) than before. In other words, I support the novelty in this part. However, the crux of the matter is that such a method indeed limits its widespread applicability. Since the author claims that this technique is mainly intended for high-quality suspended 2D, and explains its potential uses in basic 2D material properties exploration and for TEM purposes, the lack of demonstration of these target applications in the article to persuade readers that this improvement has significant benefits in these applications is a shortcoming.

Response:

We appreciate the reviewer very much for your recognition and comprehensive evaluation of our work. Using CD as the transfer medium, we developed a method for fabricating high-intactness and clean suspended monolayer graphene with 36 μm maximum suspended size achieved.

Suspended graphene film with negligible background noise is an ideal specimen support in high-resolution TEM and cryo-EM for its clean sp^2 bonded hexagonal lattice, ultrahigh thermal/electrical conductivity and mechanical strength. This point is fully proved in our previous works, including Refs [13-15] (*Adv. Mater.* 29, 1700639, 2017, *Nat. Commun.* 11, 541, 2020, *Nat. Methods* 20, 123-130, 2023, and other works, such as Ref [12] (*Science* 336, 61-64, 2012, Ref [16] (*J. Am. Chem. Soc.* 145, 8073-8081, 2023). Related content was described in the introduction part of this manuscript, as suggested by the reviewer. Importantly, larger suspended graphene area achieved in this work, enabling the facile detection of larger-sized sample with sufficiently high density.

Especially, the application of graphene-based TEM grid in cyro-electron microscopy requires the clean and high thermal conductivity of the as-fabricated suspended graphene. In this regard, thermal conductivity measurement under large suspended size

was conducted, which confirms that the suspended nature and clean surface contribute to an observed high conductivity. In addition, the fabrication of other suspended 2D materials using CD was also demonstrated in this work, and we have successfully transferred mechanically exfoliated MoS₂ and CVD-grown MoS₂ films onto TEM holey substrate, providing a strategy for investigating the intrinsic properties of 2D materials.

According to the reviewer's suggestion, related discussion on the suspended 2D materials have been added into the new version according to the reviewer's suggestion.

The revised part in manuscript,

Furthermore, suspended graphene film with perfect hexagonal lattice, ultrahigh thermal/electrical conductivity, mechanical strength and negligible background noise has been proven to have great potentials in high-resolution transmission electron microscopy (TEM) and cryo-EM imaging which enable the construction of liquid cells^{12,13}, and atomic resolution of proteins¹⁴⁻¹⁷.

References

- [12] Yuk, J. M. *et al.* High-Resolution EM of Colloidal Nanocrystal Growth Using Graphene Liquid Cells. *Science* **336**, 61-64 (2012).
- [13] Zhang, J. *et al.* Clean Transfer of Large Graphene Single Crystals for High-Intactness Suspended Membranes and Liquid Cells. *Adv. Mater.* **29**, 1700639 (2017).
- [14] Zheng, L. *et al.* Robust ultraclean atomically thin membranes for atomic-resolution electron microscopy. *Nat. Commun.* **11**, 541 (2020).
- [15] Zheng, L. *et al.* Uniform thin ice on ultraflat graphene for high-resolution cryo-EM. *Nat. Methods* **20**, 123-130 (2023).
- [16] Cheng, H. *et al.* Dual-Affinity Graphene Sheets for High-Resolution Cryo-Electron Microscopy. *J. Am. Chem. Soc.* **145**, 8073-8081 (2023).
- [17] Ahn, E. *et al.* Batch Production of High-Quality Graphene Grids for Cryo-EM: Cryo-EM Structure of *Methylococcus capsulatus* Soluble Methane Monooxygenase Hydroxylase. *ACS Nano* **17**, 6011-6022 (2023).

Question:

Furthermore, regarding wrinkle formation, I acknowledge that the author has mentioned various complex factors, but I am curious to know whether this transfer method has indeed resulted in uniform material properties, especially to convince

readers of the method's reliability. If the author believes SEM is more reflective of the morphology of suspended graphene, could an array sampling approach be used to highlight that these defects exist but are at least uniform over the whole suspended region?

Response:

Thank the reviewer for the valuable question. To characterize the uniformity of material properties, Raman spectra of single-layer suspended graphene were collected and presented in Supplementary Fig. 15, which displays no observable defect-related D peak.

Supplementary Fig. 15. Raman spectra of single-layer suspended graphene transferred with CD supporting layer.

Raman mapping result in Supplementary Fig. 16 also shows no observable D peaks for the fabricated suspended graphene in 110×110 μm² area, proving the high quality and quite high uniformity of the suspended region across the large-area region.

Supplementary Fig. 16. Raman characterization for suspended single-layer graphene. (a) The SEM image of suspended single-layer graphene on TEM SiN graphene with 5 μm diameter for each hole, the selected area is a 11×11 array with full intact graphene films. (b) Raman results for the total 121 points in the array, and there is no observable D peak. (c) Raman mapping and (d) statistic result of half-maximum of 2D peak ($\text{FWHM}_{2\text{D}}$) of suspended graphene at the selected area in (a).

According to the reviewer's suggestion, additional SEM images have been provided in the new version of SI, which can reflect the uniformity of cleanness and intactness for different suspended areas. We found that with large suspended size of graphene in single hole, for instance larger than 20 μm , graphene wrinkle would be formed owing to tendency to reduce the surface energy without any supporting medium. In contrast, with smaller hole size, almost no wrinkle would be formed (Figure R1).

Figure R1. SEM image for suspended graphene. (a-b) Some typical SEM results and no observable wrinkles can be found for small hole size. **(c-d)** Observable wrinkles can be found at large suspended area, which is roughly larger than 20 μm.

The wrinkle density for large suspended area is calculated as shown in Supplementary Fig. 11, which is varied between $0.35/\mu\text{m}$ to $0.62/\mu\text{m}$ (length of entire wrinkles to the suspended areas). Based on above observation, we believe that the wrinkles are formed owing to the releasing of the strains. With smaller hole size, our transfer method can ensure the uniformity of suspended graphene as confirmed by Raman and absence of the wrinkles. According to the reviewer's suggestion, we have included the new SEM results into the new version of supplementary information, please refer to the following discussion.

The revised part in manuscript,

*Interestingly, with large suspended, graphene wrinkle would be formed and is clearly observed from the SEM images in **Supplementary Fig. 11**, which is presumably resulted from the tendency to reduce the surface energy. In contrast, with smaller hole size,*

almost no wrinkle would be formed.

Supplementary Fig. 11. SEM images for single-layer suspended graphene and the calculated wrinkle density. (a-d), Some typical SEM images for single-layer suspended graphene with large suspended area, and the wrinkle density is calculated according to the ratio length of entire wrinkles to the suspended areas.

Question:

If the goal is to use it for fundamental 2D material property research or practical TEM cell applications, these wrinkle structures will inevitably cause an impact. Without this discussion, the current findings can only suggest that this is a method for high-yield transfer of 2D materials onto suspended substrates.

Response:

Thank the reviewer for raising this valuable comment. It is really a truth that wrinkles pose impact on properties of two-dimensional materials. Wrinkle, is a common structure in CVD-grown graphene, which is usually generated in the high-temperature process and can also be generated in transfer process. As mentioned in the reply to the last comment, with larger suspended graphene region, there would be high probability for the formation of wrinkles, owing to the tendency to release the strain. In contrast, with smaller hole size, almost no wrinkle would be formed. Furthermore, for the fabricated suspended MoS₂, either mechanically exfoliated or CVD-grown ones, there

are almost no generated wrinkles observed (Fig. 5c and Supplementary Fig. 21).

Fig. 5 Fabrication of suspended MoS₂ films. (c) SEM image of the suspended MoS₂ flakes on TEM holey grid.

Supplementary Fig. 21. SEM images for suspended MoS₂ on TEM lacey grid transferring from CVD-grown MoS₂. (a) and (b) are the SEM images of suspended MoS₂ in different magnifications with discontinuous triangular morphology shown.

Question:

Q2: The carrier film CD seems to be rigid, but when it is attached to the TEM grid, how is conformal contact between the graphene and the grid CD? If the removal of the

CD is based on sublimation without a liquid phase (as shown in the supporting video), it seems difficult to maintain perfect interfacial contact. As depicted in Supporting Figure 1b, there are stresses present during the etching of Cu in a liquid phase and during the transfer process, which prevent tight adherence of the graphene to the TEM grid. Can the mechanism of directly sublimating the CD to remove it adequately explain this?

Response:

Thank the reviewer for raising this detailed question. Supplementary Video 1 displays the quite good spreading and wetting properties of the melted CD on graphene film, and thus ensuring the good conformal contact between graphene and melting CD layer. During transfer process (Fig. 1a), the melted liquid CD (heating at 70°C to melt CD) was dropped onto TEM grid/graphene to achieve conformal contact. and the contact between graphene and CD would retain conformal contact after CD returned to solid state at room temperature, possibly owing to that the liquid-state CD would copy the surface structure of graphene on Cu (the melting point of CD is about 60 °C). And the removal of CD by its sublimation is a gradual and relatively slow process. Usually 6 h was required to completely remove CD at 40°C under air pressure, ensuring enough time for etching and cleaning.

Fig. 1 High-intactness and clean transfer method for single-layer suspended graphene with CD as support layer. (a) Schematic illustration of the procedure for transferring graphene onto TEM holey substrate.

In addition, Supplementary Figure 1b displays the sample image after complete etching of Cu substrate and cleaning. In Figure 1b, the blue thing is the image background rather than the liquid. We are sorry for the incurred misunderstanding (we have updated the captions). The sample is CD/TEM grid/graphene clamped by tweezers at the edge of CD. From the figure, we can clearly see that CD is still in solid state after etching and cleaning. This figure also demonstrates the strong mechanical property of solid CD layer that can sufficiently support graphene and entire TEM array grids.

With the conformal contact between CD and graphene, the strong mechanical property and the gradual sublimation process of CD, this transfer method can prevent potential cracks, which are usually produced by the stress from liquid solution during the transfer process. According to reviewer's concern, we have included new discussions.

The revised part in manuscript,

In the transferring, 6 h was required to completely remove CD at 40°C under air pressure at room temperature. In addition, *the strong mechanical properties of CD layer can sufficiently support graphene and the entire TEM array grids, enabling the transfer of large-area graphene onto over ten TEM grids at the same time to improve the production capacity (Supplementary Fig. 1)*, which is key to the commercial applications.

Supplementary Fig. 1. Demonstration of suspended graphene batch transferred onto holey substrate with cyclododecane supported layer. (a) The photograph of cyclododecane (CD) which fully covers holey TEM grids, below the TEM grid, it is graphene and Cu substrate in order. We can hardly identify CD because it is transparent. **(b)** *The image of CD/TEM grid/graphene clamped by tweezers after completely etching Cu substrate and washing. The material fully covers the TEM grids and in light white color is CD, and the blue part is the image background.*

Question:

Q3: In Figure S20, the CVD-grown MoS₂ appears to be a non-continuous film, showing polycrystalline triangular domains. If so, the additional description can be provided to clarify this observation.

Response:

Thank the reviewer for raising this valuable suggestion. The CVD-grown MoS₂ is truly a non-continuous film before transfer, related description has been added into the corresponding part (the original Supplementary Fig. 20 is updated as Supplementary Fig. 21 in the new version) according to the reviewer's suggestion.

The revised part,

Besides, for the fabricated suspended MoS₂ transferring from sapphire grown substrate, SEM image in **Supplementary Fig. 21** displays large discontinuous films composed of triangular morphologies of MoS₂ domains, which also proves the high-intactness transfer with the assistance of CD.

Supplementary Fig. 21. SEM images for suspended MoS₂ on TEM lacey grid transferring from CVD-grown MoS₂. (a) and (b) are the SEM images of suspended MoS₂ in different magnifications with discontinuous triangular morphology shown.

Question:

Q4: I understand that measuring mobility using the current structure might pose challenges for conducting FET or Hall measurements (though not impossible—it merely requires, substrate fabrication and pre-layout of four electrodes before transfer).

Response:

We thank the reviewer for this valuable comment, and we greatly appreciate the device array fabrication method suggested by the reviewer. However, in our suspended graphene fabrication based on CD medium, it is really a big challenge to fabricate device array for the following problems:

1. To contact the graphene, the electrodes should be deposited on patterned graphene on Cu substrate before the transfer, and the next process is to attach TEM grid onto graphene, followed by the coating of CD. However, the presence of electrode would hinder the contact between TEM grid and graphene, which would result in the cracked graphene.
2. To precisely locate the suspended regions determined by the holes of TEM grids at the pattern graphene regions is very difficult. Actually, it is very hard to ensure the good alignment between each suspended area and TEM hole in the transfer process without any damage to the TEM grid.
3. The TEM grid is very thin and can be easily destroyed by external force, the metal needle used to conduct the electrodes may damage the grid structure when the needle is used.

Therefore, we believe the suspended 2D materials fabrication method proposed in this work is not applicable for fabricating suspended graphene devices for measuring the carrier mobility. We have supplemented the Raman results to reflect the carrier mobility. We sincerely hope the reviewer can understand this.

Question:

In the supplied Figure S15, a 2D mapping is shown where the width of the spectral features varies between 23.5 to 24 cm^{-1} across different areas. How can one argue that this variation is uniform? Additionally, the cited literature uses this data to correlate with mobility. Would converting this mapping into a mobility distribution provide more compelling evidence of low defects and maintained uniformity across a large area?

Response:

We thank the reviewer for this good question. For graphene, FWHM of 2D peak ($\text{FHWM}_{2\text{D}}$) can be used to reflect the carrier mobility of device. Generally, smaller $\text{FHWM}_{2\text{D}}$ corresponds to a higher carrier mobility, and smaller variation of $\text{FHWM}_{2\text{D}}$ across large scale illustrates higher uniformity. The variation of $\text{FHWM}_{2\text{D}}$ below 10% (for supported mechanically exfoliated graphene) is regarded as uniform in the cited literature (Ref [38], *Nano Letters* 2009, 9, 2873-2876), so it is reasonable to regard the variation $22.6 \pm 0.5 \text{ cm}^{-1}$ from Raman mapping as fine uniformity.

In addition, according to the reviewer's concern, we list the comparison of $\text{FHWM}_{2\text{D}}$

mapping variation from previously reported suspended graphene. It is obvious that our work displays better $\text{FWHM}_{2\text{D}}$ results with smaller variation.

Supplementary Table 1. Comparison on $\text{FWHM}_{2\text{D}}$ of graphene from Raman mapping results using different transfer methods.

Substrates	Transfer method	Average $\text{FWHM}_{2\text{D}}$ (cm^{-1})	Range distribution of $\text{FWHM}_{2\text{D}}$ (cm^{-1})	Reference
Suspended	No-polymer transfer	24	15-33	Adv. Mater. 29, 1700639 (2017)
	PMMA-assisted	31	25-50	
	CD-assisted	22.6	21.5-24	This work
SiO_2/Si	Direct delamination of supporting films	24.3	23-26	Nat. Commun. 13, 4409 (2022)
	Freezing the medium based transfer	25.8	24-27	Adv. Mater. 36, 2308950, (2024)
	PVA-water based transfer	/	23-29	Chem. Mater. 31, 2328–2336 (2019)
	oxidation and PVA based transfer	36.5	33-40	Carbon 117 75e81, (2017)
	Triboelectricity-assisted transfer	/	32-44	Nano Res. 9, 899–907 (2016)
	PC transfer	31.8	30-44	Nanotechnol. 26, 055302 (2015)

For carrier mobility, the cited Ref [38] reported the relationship between $\text{FWHM}_{2\text{D}}$ and carrier mobility for supported graphene, which is described by the following function,

$$\mu = 1.3 \times 10^6 \times e^{-0.16\text{FWHM}_{2\text{D}}}$$

According to this relationship, the average carrier mobility is calculated as 34957 $\text{cm}^2/(\text{V s})$ in this work with 22.6 average $\text{FWHM}_{2\text{D}}$ at room temperature. The

mobility in this work is higher together with lower variation compared with that of suspended graphene transferring PMMA-assisted and buffered oxide etch ($10000 \text{ cm}^2/(\text{V s})$, *ACS Appl. Mater. Interfaces* 2023, 15, 37756–37763), and supported graphene by direct delamination of supporting films ($8800 \text{ cm}^2/(\text{V s})$, *Nat. Commun.* 2022, 13, 4409), PAN-assisted and encapsulated transfer ($11018 \text{ cm}^2/(\text{V s})$, *Adv. Mater.* 2024, 2402000).

The calculated carrier mobility for 121 Raman mapping sample is displayed in Figure R2. Of course, for the actual carrier mobility of monolayer suspended graphene in this work, some corrections on the function are needed to obtain the accurate value of carrier mobility.

Figure R2. The calculated carrier mobility for 121 number suspended graphene samples. Carrier mobility is calculated by the function in Ref [38], *Nano Letters* 9, 2873-2876 (2009).

Question:

In Figure 13, how is the ratio of residue calculated, and is this residue from the CD or other sources? It needs to be defined first.

Response:

We thank the reviewer for detailed suggestions. The impurity residue is around 1.2% by calculating the area ratio of impurity-occupied region to the entire imaged suspended graphene films (Supplementary Fig. 14, based on the pixel ratio occupied by each part in the corresponding image. And the pixel value is measured through the image processing software.

Supplementary Fig. 14. Statistics on impurity residue of graphene films on Si wafer transferred by CD. Based on AFM images, the impurity ratio is calculated by the observed impurity area to the entire characterized graphene films area.

The statistical result of impurity residue indicates the presence of very low-level residue, and there is no other element including Cu element or Na element from the XPS result, which is below the detection limitation of XPS. CD has the little chance to be the residue for its sublimation feature. The source of impurity residue may origin from residual copper oxide and airborne contamination.

Supplementary Fig. 12. X-ray photoelectron spectroscopy (XPS) characterization. (a) XPS survey spectra for pure Si wafer and the transferred graphene on Si wafer. (b-c) Enlarged XPS spectrums to detect Na and Cu elements, respectively.

According to reviewer's concern, we have included the description into the new version.

From some typical AFM images (Supplementary Fig. 13), *the impurity residue is around 1.2% by calculating the area ratio of impurity-occupied region to the entire graphene films (Supplementary Fig. 14), which may origin from residual copper oxide and airborne contaminations.*

Question:

Moreover, could the authors specify the maximum size of the suspended region achievable with this method? At what size does the yield begin to drop below 80% (or even less)?

Response:

In this work, we achieve maximum suspended size of 36 μm , which is displayed in Fig.1f. The size was only determined by the limitation of the quite little commercially available TEM grid with hole size larger than 36 μm . For calculating the intactness at large suspended size, the number of the hole in one mesh in present commercial TEM lacy grids with suspended diameter larger than 15 μm is commonly very limited (see Supplementary Figs.9-10), and the statistical result is not enough to accurately calculate and reflect the intactness at larger suspended area. Therefore, the statistic results of intactness of suspended area below 15 μm is given only. According to the reviewer's suggestion, we have included the above discussion in the new version.

The revised part in manuscript,

Moreover, the intactness of free-standing graphene as function of suspended size below

15 μm is carefully calculated, with corresponding statistic presented in **Fig. 1g**. For the suspended diameter larger than 15 μm , the statistical intactness result cannot be calculated accurately because the corresponding total number of the large hole in one grid in commercially available TEM lacy grids is commonly very limited.

Fig. 1 High-intactness and clean transfer method for single-layer suspended graphene with CD as support layer. (f) The suspended diameter for the single-layer suspended graphene is about 36 μm . **(g)** Intactness at different suspended diameters for single-layer suspended graphene obtained in this work.

Supplementary Fig. 9. SEM images for single-layer suspended graphene membranes on TEM lacy grid with large diameters in the size distribution from 20 μm to 30 μm . (a-k) Some typical SEM images for the transferred single-layer suspended graphene membranes which are intact in the size from 20 μm to 30 μm . **(i)**

The magnification image for the circled area in **(k)**, and the size of the suspended area is 27 μm . *The statistical intactness result at the suspended area larger than 15 μm cannot be accurately calculated for the number of the hole in one mesh in present commercial TEM lacy grids is commonly not enough.*

Supplementary Fig. 10. SEM images for single-layer suspended graphene on TEM lacy grid with the size of large holes in the distribution range from 30 μm to 40 μm . **(a-i)** Some typical SEM images for the transferred single-layer suspended graphene. *Note that the diameter is quite large and the hole occupies the most half of the square mesh, making it hard to calculate the intactness.* **(h)** and **(i)** are the magnification images for the circled area in **(g)** and **(k)**, respectively. We can clearly observe the large suspended graphene membranes with folds shown.

REVIEWERS' COMMENTS

Reviewer #3 (Remarks to the Author):

The author has addressed the questions raised by the referee. The production yield rate, as mentioned in Q3, can be verified by transferring onto a pre-designed hole array on a SiN substrate (like author work on 5 and 20 μm hole). I understand that this may require additional time to customize the substrate. Overall, this work is practical for application in TEM cells. Therefore, I believe this manuscript is suitable for acceptance and publication now.